# Learning Transferrable and interpretable representation for brain network

## Abstract

The human brain is a complex, dynamic network, which is commonly studied using functional magnetic resonance imaging (fMRI) and modeled as network of Regions of interest (ROIs) for understanding various brain functions. Recent studies predominantly utilize Graph Neural Networks (GNNs) to learn the brain network representation based on the functional connectivity (FC) profile, typically falling into two main categories. The Fixed-FC approaches, utilize the FC profile which represents the linear temporal relation within the brain network, is limited by failing to capture the informative temporal dynamics of brain activity. On the other hand, the Dynamic-FC approaches, modeling the evolving FC profile over time, often exhibit less satisfactory performance due to challenges in handling the inherent noisy nature of fMRI data. In this study, to address these challenges, we propose Brain Masked Auto-Encoder (BrainMAE) for learning representations directly from fMRI time-series data. Our approach incorporates two essential components—an embedding-informed graph attention mechanism and a self-supervised masked autoencoding framework. These components empower our model to capture the rich temporal dynamics of brain activity while maintaining resilience to the inherent noise in fMRI data. Our experiments demonstrate that BrainMAE consistently outperforms several established baseline models by a significant margin in three distinct downstream tasks. Finally, leveraging the model's inherent interpretability, our analysis of model-generated representations reveals intriguing findings that resonate with ongoing research in the field of neuroscience.

## 1 Introduction

Functional magnetic resonance imaging (fMRI) is a non-invasive neuroimaging technique used to measure brain activity. Due to its high spatial and temporal resolution, fMRI has become a cornerstone of neuroscience research, enabling the study of brain functions (Bullmore & Sporns, 2009; Bassett & Sporns, 2017). A general practice is to extract some brain region-of-interests (ROIs) and conceptualize the brain as a network composed of these ROIs. The connectivity between these ROIs is defined based on their linear temporal relationships, i.e. the correlation between the ROI signals. This profile of functional connectivity (FC) serves as a valuable biomarker, offering insights into the study of brain diseases (Greicius, 2008; Wang et al., 2007), aging (Ferreira & Busatto, 2013; Dennis & Thompson, 2014), and behaviors (Smith et al., 2015), and has emerged as a key tool for understanding brain function.

Recent advances have sought to leverage the rich information contained in fMRI data by applying deep learning techniques, capitalizing on their capacity for high-level representation learning. The prevalent approach in this domain involves employing Graph Neural Networks (GNNs) to extract intricate brain network representations, which can then be applied to tasks such as decoding human traits or diagnosing diseases (Kan et al., 2022b; Kawahara et al., 2017; Kim et al., 2021). These GNN models can be broadly classified into two categories based on their treatment of temporal dynamics within the data. The first category, referred to as Fixed-FC models, relies on FC matrices computed from the entire time series of fMRI data. In contrast, the second category, known as Dynamic-FC models, takes into account the temporal evolution of brain networks. These models compute FC within temporal windows using a sliding-window approach or directly learn FC patterns from the

time-series data (Kan et al., 2022a). However, both types of models exhibit certain limitations, primarily due to the unique characteristics of fMRI data.

For Fixed-FC models, depending solely on the FC profile can limit their representational capacity, as it overlooks the valuable temporal dynamics inherent in brain activities. These dynamics are often considered essential for capturing the brain's evolving states, and failing to account for them results in suboptimal brain network representations (Hutchison et al., 2013; Preti et al., 2017; Liu et al., 2018). However, current Dynamic-FC models that consider dynamic properties, often underperform Fixed-FC models (Kim et al., 2021). This discrepancy can be attributed to the intrinsic noisy nature of fMRI signals, as modeling temporal dynamics may amplify noise to some extent, whereas Fixed-FC approaches tend to mitigate noise by summarizing FC matrices using the entire time series. Furthermore, FC has been shown to be sensitive to denoising preprocessing pipelines in neuroscience studies, potentially limiting the generalizability of model representations to differently preprocessed fMRI data (Parkes et al., 2018; Li et al., 2019; Van Dijk et al., 2012).

In response to these challenges, we propose Brain Masked Auto-Encoder (BrainMAE), a novel approach for learning representations from fMRI data. Our approach effectively captures the rich temporal dynamics present in fMRI data and mitigates the impact of inherent noise through two essential components. First, drawing inspiration from the practice of word embeddings in natural language processing (NLP) (Devlin et al., 2018), we maintain an embedding vector for each ROI. These ROI embeddings are learned globally using fMRI data from all individuals in the dataset, enabling us to obtain rich representations of each ROI. We introduce an embedding-informed attention mechanism, adhering to the functionally constrained nature of brain networks, thus providing a valuable constraint in feature learning. Second, we fully leverage the information contained within the fMRI data by introducing a novel pretraining framework inspired by the concept of masked autoencoding in NLP and computer vision research (Brown et al., 2020; He et al., 2022). This masked autoencoding approach empowers our model to acquire genuine and transferable representations of fMRI time-series data. By integrating these two components, our model consistently outperforms existing models by a significant margin across several distinct downstream tasks. Furthermore, owing to its transformer-based design and inclusion of temporal components, our model provides interpretable results, shedding light on the insights it learns from the data. Lastly, we evaluate the model-generated representations, revealing intriguing findings that align with ongoing research in the field of neuroscience.

## 2 APPROACH

Our approach aims for learning the representation of fMRI data, incorporating two essential components: an embedding-informed graph attention mechanism and a masked autoencoding pretraining framework.

### 2.1 EMBEDDING-INFORMED GRAPH ATTENTION

We motivate our embedding-informed attention module based on the inherent characteristics of brain ROIs.

- **Functional Specificity.** The brain is fundamentally organized as a distributed system, with each distinct brain region serving a specific and well-defined role in the overall functioning of the brain (Power et al., 2011).
- **Functional Connectivity.** Different brain regions often are interconnected and collaborate to facilitate complex cognitive functions (Bassett & Sporns, 2017).
- **Inter-Individual Consistency.** Brain regions are known to exhibit a relatively consistent functional profile across different individuals. For instance, the primary visual cortex consistently processes visual information in nearly all individuals (Kliemann et al., 2019).

**ROI Embeddings.** There is a notable similarity in the representation properties between brain ROIs and words in natural language. Both ROIs and words possess specific functional meanings, and when combined into brain networks or sentences, they represent more complicated concepts. Furthermore, the functions of ROIs and words typically exhibit relative stability among different individuals or across sentences.

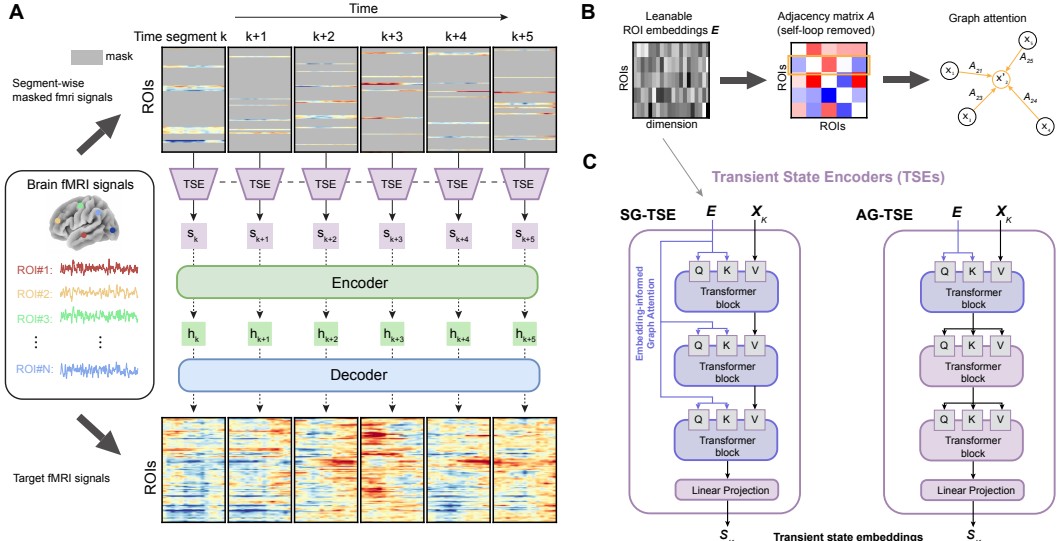

Figure 1: Schematic Overview of the Proposed BrainMAE Method. (A). Overall pre-training procedures for BrainMAE. The fMRI ROI signals are temporally segmented and, within each segment, signals of a random subset of ROIs are masked out. Masked fMRI segments are individually encoded with Transient State Encoder (TSE) and fed into a Transformer-based encoder-decoder for reconstruction. (B). ROI Embeddings-Informed Graph Attention. (C). Architecture of proposed TSE modules.

Therefore, motivated by language modeling research, we assign a learnable $d$-dimensional vector, referred to as ROI embedding, to each brain ROI. Then $N$ ROIs that cover the entire brain cortical regions form an embedding matrix denoted as $\boldsymbol{E} \in \mathbb{R}^{N \times d}$.

**Embedding-Informed Graph Attention Module.** Brain functions as a network, with brain ROIs are essentially interconnected to form a functional graph. Within this graph, each ROI is considered a node, with its node feature represented by the ROI's time-series signal of length $\tau$ or $\boldsymbol{x} \in \mathbb{R}^{\tau}$. The set of nodes in the graph is denoted as $\mathcal{V}$.

Brain ROI activities are intrinsically governed by both structural and functional networks. ROIs that are functionally confined tend to collaborate and exhibit synchronized activities (Bassett & Sporns, 2017). Drawing from these biological insights, and considering that functional relevance between ROIs can be quantified by the similarity in embeddings, we define the embedding-based graph adjacency matrix $\boldsymbol{A} \in \mathbb{R}^{N \times N}$. As illustrated in Figure 1B, each entry contains the edge weight between nodes $i, j \in \mathcal{V}$:

$$\boldsymbol{A}_{ij} = s(\boldsymbol{W}_q \boldsymbol{e}_i, \boldsymbol{W}_k \boldsymbol{e}_j). \tag{1}$$

In this equation, $s : \mathbb{R}^d \times \mathbb{R}^d \longrightarrow \mathbb{R}$ is the similarity measurement of two vectors, e.g., scaled dot-product and $\boldsymbol{e}_i, \boldsymbol{e}_j$ are the embedding vectors for node $i$ and $j$ respectively. The weight matrices $\boldsymbol{W}_q, \boldsymbol{W}_k \in \mathbb{R}^{d \times d}$ are learnable and introduce asymmetry into $\boldsymbol{A}$, representing the asymmetric information flow between two ROIs. Then, adopting the idea of graph attention mechanism, we derive the attention weight between node $i$ and node $j \in \mathcal{V} \backslash \{i\}$ as:

$$\alpha_{ij} = \mathrm{softmax}_j(\boldsymbol{A}_i) = \frac{\exp(\boldsymbol{A}_{ij})}{\sum_{k \in \mathcal{V} \backslash \{i\}} \exp(\boldsymbol{A}_{ik})}. \tag{2}$$

Grounded in the synchronized nature among functionally relevant ROIs, self-loops are removed from the attention. This design prevents the attention from favoring its own node, enhancing the reliability of the learned node features by reducing their susceptibility to input noise. Hence the feature generated by the attention module for node $i$ is

$$\boldsymbol{x}_i' = \sum_{j \in \mathcal{V} \backslash \{i\}} \alpha_{ij} \boldsymbol{x}_j \tag{3}$$

where $\boldsymbol{x}_j$ stands for the input node feature for node $j$.

In our implementation, we incorporate the embedding-informed graph attention mechanism into the standard transformer block (Vaswani et al., 2017) by simply utilizing ROI embeddings as both key and query, while the input node feature is treated as the value.

## 2.2 BRAIN MASKED AUTOENCODER

In order to effectively capture the temporal dynamics and extract the genuine representation from fMRI signals, we utilize a transformer-based encoder-decoder architecture and design a nontrivial self-supervised task for pretraining, as shown in Figure 1A.

**Temporal Segmentation.** Similar to previous studies on vision transformers, where 2D images are divided into non-overlapping patches (He et al., 2022), in this context, we temporally segment the fMRI signals. Each segment has the shape of $N \times \tau$, where $\tau$ represents the length of each time window. Such segmentation allows transformer-like models to be seamlessly applied, as each fMRI segment can be viewed as a token, and thus the original fMRI data can be represented as a sequence of tokens. Throughout our study, the length of the segment $\tau$ is set to 15 seconds aligning with the typical duration of transient events in fMRI data (Shine et al., 2016; Bolt et al., 2022).

**Transient State Encoders (TSEs).** We embed each fMRI segment denoted as $\mathbf{X}_k \in \mathbb{R}^{N \times \tau}$ using Transient State Encoders (TSEs), the architectural details of which are illustrated in Figure 1C. We introduce two types of TSEs, namely Static-Graph TSE (**SG-TSE**) and Adaptive-Graph TSE (**AG-TSE**), which indicate how node features are learned.

Both TSEs consist of three transformer blocks but differ in the attention mechanisms applied within these layers. For SG-TSE, all three transformer blocks exclusively employ embedding-informed graph attention mechanism, assuming "static" connectivity among brain regions. On the other hand, AG-TSE incorporates two self-attention blocks stacked on top of an embedding-informed attention block, allowing the attention to be "adaptive" to the node input signal, enabling the model to capture the transient reorganization of brain connectivity. The output from the transformer blocks forms a matrix $\boldsymbol{X}^o \in \mathbb{R}^{N \times d}$, where each row represents the feature learned for each ROI. We employ a linear projection $g : \mathbb{R}^{N \times d} \longrightarrow \mathbb{R}^d$ to aggregate all of the ROI features into a single vector $s_k \in \mathbb{R}^d$. This vector, as the output of TSE, represents the transient state of fMRI segment $k$.

**Segment-wise ROI Masking.** Different from the masked autoencoding commonly used in image or language modeling studies (He et al., 2022; Devlin et al., 2018), where tokens or image patches are typically masked out, we employ a segment-wise ROI masking approach. Specifically, for each fMRI segment, we randomly choose a subset of the ROIs, such as 70% of the ROIs, and then mask out all of the signals within that segment from those selected ROIs. The masked ROI segments are replaced with a masked token which is a shared and learnable $d$-dimensional vector that indicates the presence of a missing ROI signal. This masking scheme introduces a nontrivial reconstruction task, effectively guiding the model to learn the functional relationships between ROIs.

**Autoencoder.** We employ a transformer-based autoencoder to capture both the temporal relationships between fMRI segments and the overall fMRI representation. The encoder maps the input sequence of transient state embeddings generated by the TSE $(\boldsymbol{s}_1, \boldsymbol{s}_2 \ldots, \boldsymbol{s}_n)$ to a sequence of hidden representations $(\boldsymbol{h}_1, \boldsymbol{h}_2, \ldots, \boldsymbol{h}_n)$. Subsequently, the decoder reconstructs the fMRI segments $(\hat{\boldsymbol{X}}_1, \hat{\boldsymbol{X}}_2, ..., \hat{\boldsymbol{X}}_n)$ based on these hidden representations. Both the encoder and decoder consist of two standard transformer blocks and position embeddings are added for all tokens in both the encoder and decoder. The decoder is only used in pre-training phase and omitted from downstream task fine-tuning.

**Reconstruction Loss.** We compute Mean Squared Error (MSE) loss to evaluate the reconstruction error for masked ROI segments and unmasked ROI segments separately:

$$\mathcal{L}_{\text{mask}} = \sum_{k=1}^{n} \frac{1}{n\tau|\Omega_k|} \sum_{i \in \Omega_k} \|\hat{\boldsymbol{X}}_{k,i} - \boldsymbol{X}_{k,i}\|^2 \tag{4}$$

$$\mathcal{L}_{\text{unmask}} = \sum_{k=1}^{n} \frac{1}{n\tau|\mathcal{V}\backslash\Omega_k|} \sum_{i \in \mathcal{V}\backslash\Omega_k} \|\hat{\boldsymbol{X}}_{k,i} - \boldsymbol{X}_{k,i}\|^2 \tag{5}$$

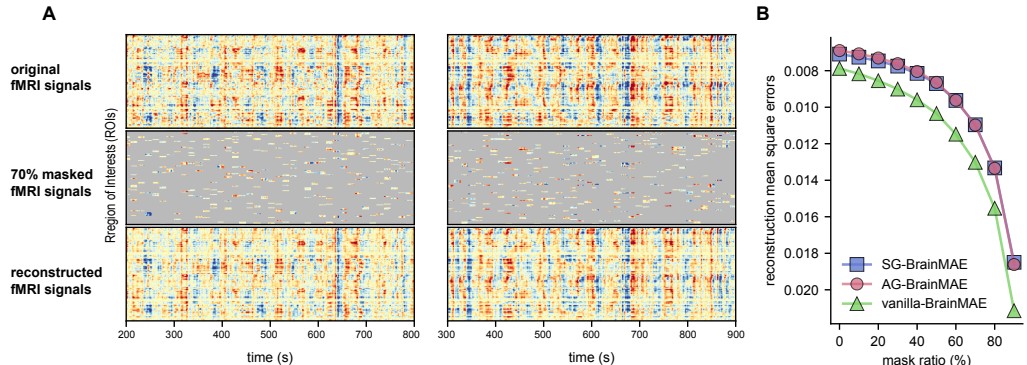

Figure 2: Reconstruction results on HCP-7T using HCP-3T pretrained model. (A) Example results. (B) Reconstruction error across mask ratios for SG-, AG-, and vanilla-BrainMAEs.

where $n$ is total number of fMRI segments, $\Omega_k$ is the set of masked ROI in the $k$-th segments, $\boldsymbol{X}_{k,i}$ is the $k$-th fMRI segment of the $i$-th ROI and $\hat{\boldsymbol{X}}_{k,i}$ is the reconstructed one. The total reconstruction loss for pretraining the model is the weighted sum of the two:

$$\mathcal{L} = \lambda \mathcal{L}_{\text{mask}} + (1 - \lambda)\mathcal{L}_{\text{unmask}} \tag{6}$$

where $\lambda$ is a hyperparameter, and in our study, we set $\lambda$ to a fixed value of 0.75.

**BrainMAEs.** Based on the choice of TSE, we introduce two variants of BrainMAE: SG-BrainMAE and AG-BrainMAE, incorporating SG-TSE and AG-TSE for transient state encoding, respectively.

## 3 EXPERIMENTS

### 3.1 DATASETS

We mainly use the following datasets to evaluate our approach. **HCP-3T** dataset (Van Essen et al., 2013) is a large-scale publicly available dataset that includes 3T fMRI data from 897 healthy adults aged between 22 and 35. We use the resting-state sessions as well as the behavior measurements in our study. **HCP-7T** dataset is a subset of the HCP S1200 release, consisting of 7T fMRI data from 184 subjects within the age range of 22 to 35. Our analysis focused on the resting-state sessions of this dataset. **HCP-Aging** dataset (Harms et al., 2018), designed for aging studies, contains 725 subjects aged 36 to 100+. Age, gender information as well as the resting-state fMRI data are used in our study. **NSD** dataset (Allen et al., 2022) is a massive 7T fMRI dataset, featuring 8 human participants, each with 30-40 scan sessions conducted over the course of a year. For our study, we incorporated both task fMRI data and task performance metrics, including task scores and response times (RT). Detailed information regarding each of the datasets can be found in Appendix C.

### 3.2 PRE-TRAINING EVALUATION

#### 3.2.1 IMPLEMENTATION DETAILS

During the pretraining phase, each time we randomly select 300 seconds of fixed-length fMRI signals from the original sample and divide this signal into 20 segments, with each segment containing 15 seconds of fMRI data. We use a variable mask ratio for each mini-batch during training. The mask ratio for each mini-batch is drawn from a range of (0, 0.8), where 0 indicates no masking is applied. For each pretrained dataset, we train the model using all available samples for 1000 epochs. Additional training settings are available in A.1.

#### 3.2.2 MASKED SIGNAL RECONSTRUCTION

In this section, we assess the reconstruction performance of the HCP-3T pretrained model on unseen HCP-7T dataset. Example reconstruction results are shown in Figure 2A and Appendix H.

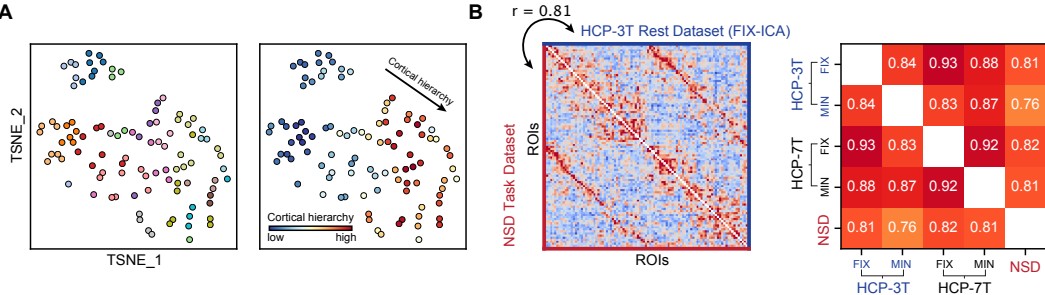

Figure 3: Evaluation of pretrained ROI embeddings. (A) A t-SNE plot of the HCP-3T pretrained ROI embeddings. Yeo-17 network (Left) and cortical hierarchy (Right) can be identified. (B) Consistency of ROI embedding similarity matrix across different datasets.

Furthermore, in Figure 2B, we conduct a comparison of the reconstruction error between the proposed models and a variant employing only self-attention within the TSE model, which we term vanilla-BrainMAE (further details in B.1). Both SG-BrainMAE and AG-BrainMAE consistently achieve lower reconstruction errors across all mask ratios. This suggests that the incorporation of the embedding-informed graph attention module is advantageous for acquiring more generalized representations.

### 3.2.3 ROI Embeddings

It is crucial to validate whether the ROI embeddings pretrained with the proposed approach truly satisfy the aforementioned ROI characteristics.

**Functional Specificity.** We visualize t-SNE transformed ROI embeddings in Figure 3A (Van der Maaten & Hinton, 2008). In the projected 2D space, the ROIs exhibit discernible clustering that aligns with the Yeo-17 networks' definitions in neuroscience studies Yeo et al. (2011). This alignment suggests that ROIs with similar functional roles display similar embeddings. In other words, region functions can be inferred from these embeddings.

**Functional Connectivity.** Interestingly, the arrangement of the ROI in the projected 2D space also reflects the cortical hierarchy, as indicated by principal gradient (PG) values (Margulies et al., 2016; Gu et al., 2021; Raut et al., 2021). Low PG values correspond to cortical low-order regions, such as visual and somatosensory regions, while high PG values correspond to cortical high-order regions, including the default mode network and limbic system. The interconnectivity between different brain networks thus can also be informed by the ROI embeddings.

**Inter-Individual Consistency.** We separately pretrain the SG-BrainMAE on the HCP-3T, HCP-7T, and NSD task datasets. Notably, both HCP-3T and HCP-7T datasets have two different preprocessing pipelines, namely minimal preprocessing and FIX-ICA. Consequently, we pretrained models for each combination. In total, we obtained five independently pretrained models. For each pretrained model, we generated an embedding similarity matrix by computing pairwise cosine similarities between ROIs, as illustrated in Figure 3B. Importantly, these similarity matrices consistently exhibit similar patterns across different datasets, regardless of preprocessing pipelines or fMRI task types (resting or task), suggesting the converging ROI representations. More ROI analysis is shown in D.

### 3.3 Transfer Learning Evaluation

### 3.3.1 Implementation Details

We pretrain the model using HCP-3T ICA-FIX preprocessed data and fine-tune the whole network except for the ROI embeddings for each downstream task. Similar to previous studies (He et al., 2022; Devlin et al., 2018), we use the CLS token from the encoder and append a task-specific linear head for prediction. Cross-entropy loss and mean squared error (MSE) loss are used for classification and regression fine-tuning tasks respectively. For fair comparisons, we use 5-fold cross-validation for model evaluation. Detailed information regarding the architecture and training settings can be found in A.2.

Table 1: Results for behavior prediction.

| Model | Gender | | Behaviors (measured in MAE) | | | | | | | |
|---|---|---|---|---|---|---|---|---|---|---|
| | Accuracy (%) | AUROC (%) | PicSeq | PMT_CR | PMT_SI | PicVocab | IWRD | ListSort | LifeSatisf | PSQI |
| *Fixed-FC* | | | | | | | | | | |
| BrainNetTF-OCR | 94.11±0.98 | 94.39±0.90 | 7.11±0.22 | 2.28±0.11 | 1.72±0.12 | 4.70±0.10 | 1.56±0.03 | 5.96±0.07 | 4.96±0.22 | 1.49±0.05 |
| BrainNetTF-Vanilla | 90.00±1.05 | 89.93±0.94 | 8.19±0.46 | 2.73±0.07 | 2.13±0.07 | 5.93±0.14 | 1.81±0.08 | 6.91±0.21 | 5.71±0.08 | 1.68±0.05 |
| BrainNetCNN | 90.68±1.80 | 90.89±1.55 | 10.21±0.22 | 3.25±0.13 | 2.64±0.14 | 6.65±0.27 | 2.26±0.05 | 8.51±0.20 | 7.12±0.24 | 1.68±0.05 |
| *Dynamic-FC* | | | | | | | | | | |
| STAGIN-SERO | 88.73±1.36 | 88.69±1.41 | 10.22±0.15 | 3.49±0.05 | 2.70±0.06 | 6.78±0.20 | 2.26±0.05 | 8.51±0.20 | 7.12±0.24 | 2.12±0.04 |
| STAGIN-GARO | 88.34±0.94 | 88.33±0.91 | 10.26±0.18 | 3.44±0.10 | 2.69±0.09 | 6.92±0.30 | 2.25±0.04 | 8.52±0.26 | 7.09±0.35 | 2.08±0.04 |
| FBNETGNN | 88.05±0.15 | 87.93±0.97 | 8.62±0.21 | 2.93±0.11 | 2.34±0.11 | 5.83±0.15 | 1.96±0.04 | 7.31±0.10 | 6.09±0.10 | 1.81±0.03 |
| *Ours* | | | | | | | | | | |
| SG-BrainMAE | **97.49±0.15** | **97.46±0.18** | **5.06±0.21** | **1.63±0.08** | **1.24±0.04** | **3.40±0.14** | **1.11±0.04** | **4.35±0.12** | 3.64±0.27 | **1.05±0.06** |
| AG-BrainMAE | 97.13±0.56 | 97.17±0.61 | 5.09±0.05 | 1.67±0.10 | 1.28±0.06 | 3.34±0.11 | 1.13±0.03 | 4.37±0.06 | **3.58±0.17** | 1.07±0.05 |

Table 2: Results for age prediction.

| Model | Gender | | Age (MAE) |
|---|---|---|---|
| | Accuracy (%) | AUROC (%) | |
| *FIX-FC* | | | |
| BrainNetTF-OCR | 90.21±3.81 | 90.73±2.85 | 6.15±0.71 |
| BrainNetTF-Vanilla | 88.96±2.16 | 88.76±2.21 | 6.78±0.56 |
| BrainNetCNN | 88.83±1.52 | 88.74±1.58 | 8.71±0.62 |
| *Dynamic-FC* | | | |
| STAGIN-SERO | 82.37±1.66 | 82.57±1.36 | 8.96±0.47 |
| STAGIN-GARO | 80.67±0.81 | 80.58±1.03 | 8.65±0.28 |
| FBNETGNN | 89.50±3.58 | 89.34±3.49 | 6.68±1.00 |
| *Ours* | | | |
| SG-BrainMAE | **92.67±1.07** | **92.51±1.07** | **5.78±0.44** |
| AG-BrainMAE | 91.12±1.99 | 91.15±2.03 | 6.49±1.00 |

Table 3: Results for task performance prediction.

| Model | Task score (MAE) | RT (MAE in ms) |
|---|---|---|
| *FIX-FC* | | |
| BrainNetTF-OCR | 0.070±0.003 | 92.344±2.343 |
| BrainNetTF-Vanilla | 0.075±0.004 | 96.252±2.133 |
| BrainNetCNN | 0.078±0.004 | 102.911±2.225 |
| *Dynamic-FC* | | |
| STAGIN-SERO | 0.089±0.003 | 116.635±2.197 |
| STAGIN-GARO | 0.091±0.002 | 116.130±2.099 |
| FBNETGNN | 0.074±0.005 | 95.349±2.320 |
| *Ours* | | |
| SG-BrainMAE | **0.069±0.004** | **90.678±1.767** |
| AG-BrainMAE | 0.070±0.003 | 92.154±2.265 |

### 3.3.2 BASELINES

In our evaluation, we benchmark our approach against two categories of baseline neural network models designed for brain network analysis.

**Models based on fixed brain network (Fixed-FC).** BrainNetTF-OCR (Kan et al., 2022b) is transformer-based model with Orthonormal Clustering Readout (OCR), representing the state-of-the-art method for brain network analysis. BrainNetTF-Vanilla is a variant of BrainNetTF with CONCAT-based readout. BrainNetCNN (Kawahara et al., 2017) follows the CNN paradigm by modeling the functional connectivity matrices similarly as 2D images.

**Models based on dynamic brain network (Dynamic-FC).** STAGIN (Kim et al., 2021) is a transformer-based model that learns dynamic graph representation with spatial-temporal attention. Two variants of the models, STAGIN-GARO and STAGIN-SERO that use different readout methods are included for comparison. FBNETGEN (Kan et al., 2022a) is a GNN-based model that learns the brain network from the fMRI time-series signals.

### 3.3.3 PERFORMANCE COMPARISON

We compare our model against baseline methods across three distinct downstream tasks.

**HCP-3T dataset: Behaviors prediction.** In this task, the models are required to simultaneously perform gender classification as well as predict 8 behavior measurements corresponding to various cognitive functions. Definitions of these behavior measurements can be found in Table 9. The models are trained/fine-tuned using HCP-3T dataset with ICA-FIX preprocessed fMRI data.

**HCP-Aging dataset: Age prediction.** For this task, the models are required to simultaneously perform gender classification and age prediction from fMRI data.

**NSD dataset: Task performance prediction.** For this task, the models are required to predict the averaged memory task score as well as the response time for each fMRI run.

Table 4: Results for age prediction

| Model | Gender | | Aging (MAE) |
|---|---|---|---|
| | Accuracy (%) | AUROC (%) | |
| SG-BrainMAE | **92.67±1.07** | **92.51±1.07** | **5.75±0.44** |
| AG-BrainMAE | 91.12±1.99 | 91.15±2.03 | 6.49±1.00 |
| vanilla-BrainMAE | 88.54±2.50 | 88.53±2.37 | 7.26±0.95 |
| vanilla-BrainAE | 80.92±2.40 | 81.03±2.52 | 8.33±0.49 |

Table 5: Results for task performance prediction

| Model | Task Score (MAE) | RT (MAE in ms) |
|---|---|---|
| SG-BrainMAE | **0.069±0.004** | **90.678±1.767** |
| AG-BrainMAE | 0.070±0.004 | 92.154±2.265 |
| vanilla-BrainMAE | 0.083±0.004 | 108.215±3.458 |
| vanilla-BrainAE | 0.091±0.004 | 118.965±3.047 |

The results for the three downstream tasks are shown in Table 1, 2, and 3 and reported as mean ± std. from 5-fold cross-validation. Our proposed methods consistently demonstrate superior performance across all these tasks. To assess the model's ability to extract genuine information that is generalizable across different fMRI preprocessing pipelines, we evaluate the model trained with HCP-3T ICA-FIX data on HCP-3T minimal preprocessed data. As results reported in Table 20, our approach outperforms other methods, indicating better generalizability in representation learning.

Notably, despite the informative nature of temporal features, the three baseline models that leverage dynamic FC or learn the FC from the fMRI time series consistently underperform compared to models that utilize fixed FC. One plausible explanation could be attributed to the inherent noise present in fMRI data. In contrast, even though our model encodes the complete temporal information of fMRI, our models still achieve the highest level of performance. This achievement can be attributed to the static-graph attention module employed in our model design.

### 3.3.4 ABLATION STUDY

We conduct ablation studies on the aforementioned three downstream tasks to elucidate the advantage of incorporating the embedding-informed graph attention module and masked autoencoding in our approach. We compare four model variants: (1) SG-BrainMAE, (2) AG-BrainMAE, (3) vanilla-BrainMAE (BrainMAE without the embedding-informed attention modules, see B.1 for details), and (4) vanilla-BrainAE (sharing the same architecture as vanilla-BrainMAE but pretrained with standard autoencoding, see B.2 for details). The differences between these variants are detailed in Table 22. The results, which are presented in Table 4, 5, and 21, indicate a degradation in performance when the embedding-informed attention is removed, and a further significant decrease in performance when masked pretraining is excluded. This underscores the advantage of incorporating both components in BrainMAE framework.

### 3.3.5 REPRESENTATION AND INTERPRETATION ANALYSIS

We evaluate the learned representation of our model fine-tuned on HCP-3T behaviors task. The fMRI representation (the features used by the task-specific head) is extracted for each individual and combined to perform principal component analysis (PCA). As shown in Figure 4A-C, we find that, despite the model is trained to predict multiple variables, two components explain a significant amount of the variance (approximately 55%). As expected, the first component predominantly represents gender, but surprisingly, the second component exhibits a strong correlation with the CCA mode (Smith et al., 2015), which resembles a 'positive-negative' axis linking lifestyle, demographic, and psychometric measures. This intriguing result suggests that the model is able to extract higher-level abstract representation that reflects individual's general behaviors and lifestyles.

We interpret our model fine-tuned on NSD task performance dataset. We evaluate self-attention scores used to generate the CLS representation of the final layers of the encoder transformer. These attention scores provide insights into which time segments are crucial for the model to make predictions. As illustrated in Figure 4D, we average the attention scores across different fMRI runs. Notably, the attention score reveals the NSD task block structure, indicating that the model inherently places more importance on task blocks to infer overall task performance. Interestingly, the attention score correlates to the behavioral arousal measurements (inverse response time (Makovac et al., 2019)) across- and within-task blocks, suggesting the model is aware of the change of transient brain state. Indeed, the learned fMRI representation also highly correlates with brain arousal index (see E.2). Overall, these results underline the versatility of our model, hinting at its potential to explore brain mechanisms for neuroscience research in a data-driven manner.

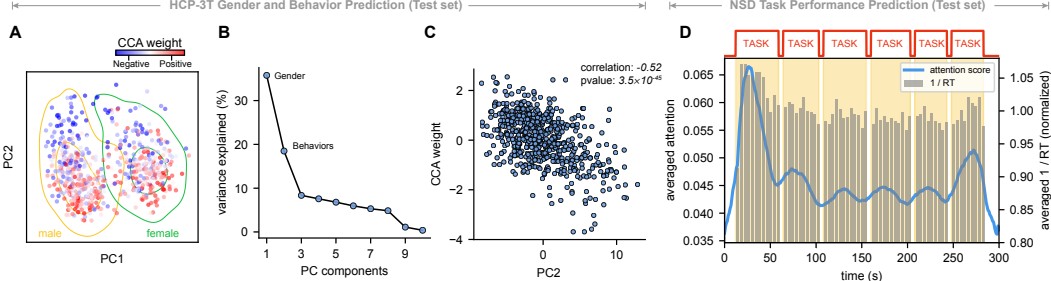

Figure 4: Model representation and interpretation. (A) The 2D plot of the top-2 principal components (PCs) of learned representation. (B) Variance explained by each PCs. (C) Strong correlation between PC2 and behavior CCA mode score. (D) Model "pays more attention" on task blocks and the attention score corresponds to brain arousal state change (measured by 1/RT).

## 4 RELATED WORK

**Masked autoencoding.** The incorporation of masked autoencoding for self-supervised representation learning has seen significant interest across various domains. In the realm of NLP, models like BERT (Devlin et al., 2018) and GPT (Brown et al., 2020; Radford et al., 2018; 2019) employ masked autoencoding to pretrain language models by predicting the missing components of input sequences. In computer vision, masked modeling has been integrated with vision transformers, yielding successful approaches such as MAE (He et al., 2022), BEiT (Bao et al., 2021), and BEVT (Wang et al., 2022) for feature learning. Limited studies explore masked autoencoding in the context of fMRI. Thomas et al. (2022) adapt BERT-like framework by considering each time point as 'word', which is suitable for modeling transient states but limited in scaling to fMRI signals of hundreds seconds. Our approach differs by treating transient state as basic unit for sequence learning, allowing it to scale effectively and extract representations that reflect individual traits and behaviors.

**Brain Network Analysis.** GNN-based models have been widely used in the field of brain network analysis. Models like GroupINN (Yan et al., 2019) introduce the concept of node grouping to enhance interpretability and reduce model size. BrainNetCNN (Kawahara et al., 2017) capitalizes on the topological locality of structural brain networks for graph-based feature learning. BrainNetTF (Kan et al., 2022b), on the other hand, is a transformer-based model that employs orthonormal clustering readout to facilitate cluster-aware graph embedding learning. STAGIN (Kim et al., 2021) focuses on learning dynamic graph representations through spatial-temporal attention mechanisms, while FBNetGen (Kan et al., 2022a) directly learns the brain network from fMRI time-series data. In contrast to these methods, which predominantly learn representation from functional connectivity (FC), our approach can effectively incorporate valuable temporal information while mitigating the impact of fMRI's intrinsic noise through specifically designed modules.

## 5 DISCUSSION AND CONCLUSION

Here we propose BrainMAE for effectively learning the representation from fMRI time series data. Our approach integrates two essential components: an embedding-informed static-graph attention module and a masked self-supervised pretraining framework. These components are designed to capture temporal dynamics while mitigating the inherent noise in fMRI data. The alignment of the learned ROI embeddings with existing neuroscience knowledge, along with the improvement in transfer learning tasks, confirms the effectiveness of our design.

By providing a task-agnostic representation, our model exhibits promise for applications in the field of neuroscience. Its interpretability and ability to capture transient representations make it a valuable tool for uncovering the mechanisms and dynamics of transient state changes within the brain.

Furthermore, our approach can be extended beyond brain fMRI data. It can be applied to various domains that can be modeled as networks of functionally meaningful nodes. For instance, it could be applied to traffic network analysis, where the nodes represent either roads or some spatially defined regions.

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

| Table 6: Pretraining Settings | |
|---|---|
| config | value |
| optimizer | AdamW |
| Training epochs | 1000 |
| weight decay | 0.05 |
| optim momentum | $\beta 1, \beta 2 = 0.9, 0.95$ |
| Base learning rate | 0.001 |
| $\lambda$ | 0.75 |
| batch size | 32 |
| batch accumulation | 4 |
| learning rate schedule | cosine decay |
| warmup epochs | 100 |

| Table 7: Fine-tuning settings | |
|---|---|
| config | value |
| optimizer | AdamW |
| Training epochs | 150 |
| Train:Val:Test (each fold) | 0.64:0.2:0.16 |
| weight decay | 0.05 |
| optim momentum | $\beta 1, \beta 2 = 0.9, 0.95$ |
| Base learning rate | 0.001 |
| batch size | 64 |
| batch accumulation | 2 |
| learning rate schedule | cosine decay |
| textclip_grad | 5 |

## A  EXPERIMENT DETAILS

### A.1  PRETRAINING

Table 6 provides a summary of our pretraining configurations, which were employed for training BrainMAE on various datasets. Our ROIs are extracted based on Schaefer2018_100ROIs Parcels, which include 100 cortical ROIs (Schaefer et al., 2018). During the training process, we employ a random selection method to choose a continuous segment of fMRI signals lasting for 300 seconds. For instance, we randomly select 300 consecutive fMRI signals from the original 864 seconds of data. This ensures that the signal used for masking and subsequent reconstruction is of equal size, allowing for the construction of mini-batches for parallelized training.

Additionally, this approach offers the advantage of efficient GPU memory utilization and scalability. It also introduces a degree of randomness, which, to some extent, serves as data augmentation and benefits representation learning.

### A.2  TRANSFER LEARNING

We fine-tune all BrainMAE models following the parameters outlined in Table 7. During both the training and testing phases, we utilize the original length of the fMRI data. For example, in the case of HCP3T with 864 seconds of data, we use the first 855 seconds, dividing it into 57 time segments that are fed into the model. The decoder is omitted during transfer learning. A task-specific linear head is appended to the CLS token representation generated by the encoder transformer for task-specific predictions, as shown in Figure 5A.

## B  MODEL VARIANTS

### B.1  VANILLA-BRAINMAE

The vanilla-BrainMAE shares the exact same architecture as BrainMAE with the only exception being the use of vanilla-TSE to extract transient state embeddings from the fMRI segments. The vanilla-TSE, as shown in Figure 5, incorporates three standard transformer blocks that exclusively utilize self-attention. The vanilla-BrainMAE serves as a model for comparison with both AG-BrainMAE and SG-BrainMAE, allowing us to evaluate the proposed embedding-based static graph module.

### B.2  VANILLA-BRAINAE

The vanilla-BrainAE employs the exact same architecture as vanilla-BrainMAE, with the only distinction being the use of traditional autoencoding for fMRI signal reconstruction without signal masking. vanilla-BrainAE is included as a comparative model to assess the proposed masked autoencoding approach in comparison to the other models.

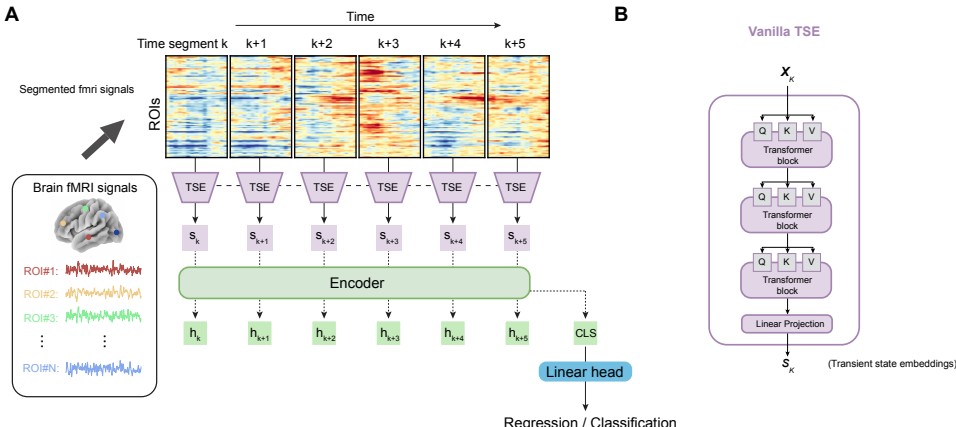

Figure 5: A. BrainMAE fine-tuning framework.B. vanilla-TSE modules utilizing only the self-attention in the transformer blocks.

## C DATASETS

**HCP-3T/HCP-7T Datasets.**

The Human Connectome Project (HCP) is a freely shared dataset from 1200 young adult (ages 22-35) subjects, using a protocol that includes structural images (T1w and T2w), functional magnetic resonance imaging (resting-state fMRI, task fMRI), and high angular resolution diffusion imaging (dMRI) at 3 Tesla (3T) and behavioral and genetic testing. Moreover, 184 subjects also have 7T MR scan data available (in addition to 3T MR scans), which includes resting-state fMRI, retinotopy fMRI, movie-watching fMRI, and dMRI. In our study, we focused on the resting-state session of the dataset as well as 8 behavior measurements (see Table 9 for more information).

**HCP-Aging Dataset.**

The Human Connectome Project Aging (HCP-Aging) dataset is an extensive and longitudinally designed neuroimaging resource aimed at comprehensively investigating the aging process within the human brain. It comprises a wide array of multimodal neuroimaging data, such as structural MRI (sMRI), resting-state functional MRI (rs-fMRI), task-based fMRI (tfMRI), and diffusion MRI (dMRI), alongside rich cognitive and behavioral assessments.In our study, we focused on the resting-state session of the dataset as well as age and gender information.

**NSD Dataset.**

The Natural Scenes Dataset comprises whole-brain 7T functional magnetic resonance imaging (fMRI) scans at a high resolution of 1.8 mm. These scans were conducted on eight meticulously selected human participants, each of whom observed between 9,000 and 10,000 colorful natural scenes, totaling 22,000 to 30,000 trials, over the span of one year. While viewing these images, subjects were engaged in a continuous recognition task in which they reported whether they had seen each given image at any point in the experiment.

Table 8: Dataset Statistics

|  | HCP-3T | HCP-7T | HCP-Aging | NSD |
|---|---|---|---|---|
| number of subjects | 897 | 184 | 725 | 8 |
| number of sessions | 3422 | 720 | 2400 | 3120 |
| Number of TRs | 1200 | 1200 | 478 | 301 |
| Orignal TR(s) | 0.72 | 1.00 | 0.80 | 1.00 |
| Number of TR interpolate to 1s | 864 |  | 382 |  |
| type | Resting-state | Resting-state | Resting-state | TASK |

Table 9: Behaviors description

| Behavior | Display Name | Description |
|---|---|---|
| PicSeq | NIH Toolbox Picture Sequence Memory Test: Unadjusted Scale Score | The Picture Sequence Memory Test is a measure developed for the assessment of episodic memory for ages 3-85 years. Participants are given credit for each adjacent pair of pictures,the maximum score is 17(because that is the number of adjacent pairs of pictures that exist). |
| PMT_CR | Penn Progressive Matrices: Number of Correct Responses (PMAT24_A_CR) | Penn Matrix Test: Number of Correct Responses. A measure of abstraction and mental flexibility. It is a multiple choice task in which the participant must conceptualize spatial, design and numerical relations that range in difficulty from very easy to increasingly complex. |
| PMT_SI | Penn Progressive Matrices: Total Skipped Items (PMAT24_A_SI) | Penn Matrix Test: Total Skipped Items (items not presented because maximum errors allowed reached). |
| PicVocab | NIH Toolbox Picture Vocabulary Test: Unadjusted Scale Score | This measure of receptive vocabulary is administered in a computerized adaptive format. The respondent is presented with an audio recording of a word and four photographic images on the computer screen and is asked to select the picture that most closely matches the meaning of the word. |
| IWRD | Penn Word Memory Test: Total Number of Correct Responses (IWRD_TOT) | Participants are shown 20 words and asked to remember them for a subsequent memory test. They are then shown 40 words (the 20 previously presented words and 20 new words matched on memory related characteristics). |
| ListSort | NIH Toolbox List Sorting Working Memory Test: Unadjusted Scale Score | This task assesses working memory and requires the participant to sequence different visually- and orally-presented stimuli. Pictures of different foods and animals are displayed with both a sound clip and written text that name the item. Participants are required to order a series of objects (either food or animals) in size order from smallest to largest. |
| LifeSatisf | NIH Toolbox General Life Satisfaction Survey: Unadjusted Scale Score | Life Satisfaction is a concept within the Psychological Well-Being subdomain of Emotion. Life Satisfaction is one's cognitive evaluation of life experiences and is concerned with whether people like their lives or not. This self-report measure is a 10-item calibrated scale comprised of items from the Satisfaction with Life Scale. |
| PSQI | Sleep (Pittsburgh Sleep Questionnaire) Total Score | The Pittsburgh Sleep Quality Index (PSQI) is a self-rated questionnaire which assesses sleep quality and disturbances over a 1-month time interval.Scores for each question range from 0 to 3, with higher scores indicating more acute sleep disturbances. |

Table 8 provides an overview of the statistical information for each of the datasets employed in our study.

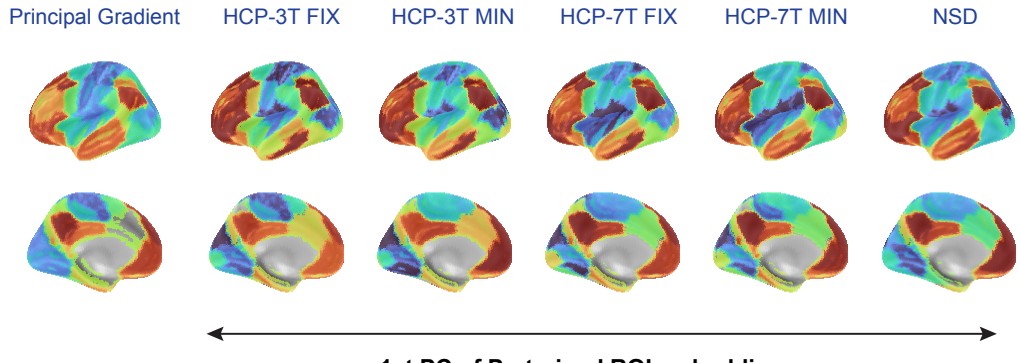

**1st PC of Pretrained ROI embeddings**

Figure 6: Comparison between principal gradient and the first principal component of pretrained ROI embeddings on different datasets. The color mapped on the brain surface encodes either principal gradient value (1st column) or 1st PC value of pretrain embeddings (2-6th columns).

## D  ROI EMBEDDING ANALYSIS

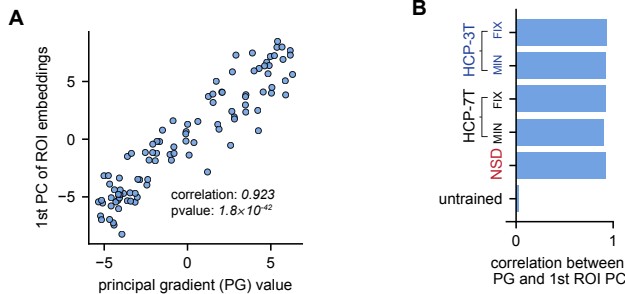

Figure 7: A. Strong correlation between 1st PC of ROI embedding (HCP-3T pretrained) and principal gradient. B. This relationship is hightly reproducible across differently pretrianed models but is not the case for untrained model.

### D.1  RELATIONSHIP TO PRINCIPAL GRADIENT

In neuroscience research, the principal gradient characterize the topographical organization of brain regions, reflecting the between network functional organization (Margulies et al., 2016). Along this principal direction, one end is associated with regions serving primary sensory/motor functions, while the other end corresponds to transmodal regions, often referred to as the default-mode network (DMN).

We have identified a significant relationship between the first principal component of pretrained ROI embeddings and the principal gradient. This finding suggests that the functional connectivity between brain networks is inherently encoded in these embeddings. Furthermore, this relationship is highly reproducible across pretrained models trained on different datasets, as shown in Figure 6 and 7.

### D.2  CONSISTENCY ACROSS PRETRAINED MODELS

We analyze the cross-region embedding similarity, or embeddings similarity matrix for each of the models pretrained on various datasets. We use the cosine distance to measure the similarity between two ROI embedding vectors. As shown in Figure 8, The embedding similarity matrices shows converging results on differently pretrained models, suggesting that highly similar embedding profiles can be identified in different datasets, thereby validating our hypothesis and the proposed approach.

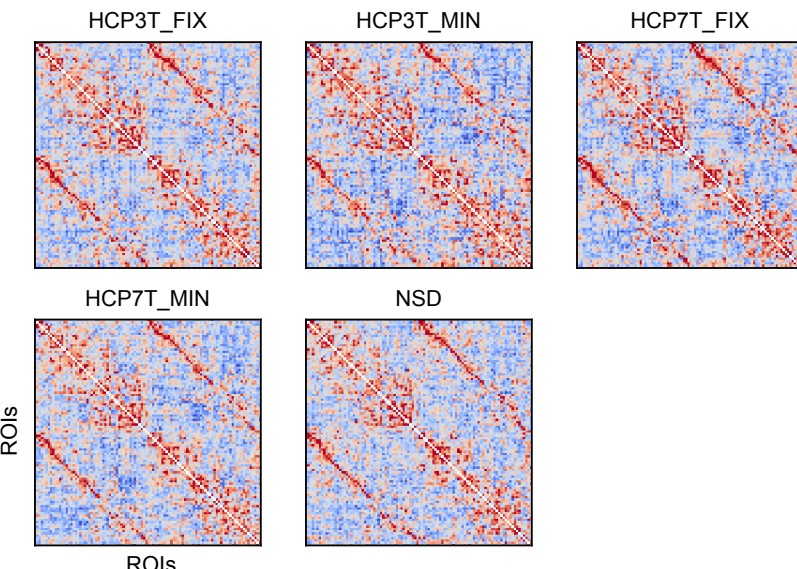

Figure 8: The convergence result observed in the cross-regional embedding similarity matrix reveals the consistency of ROI embeddings pretrained on different datasets.

## D.3  AGE EFFECTS

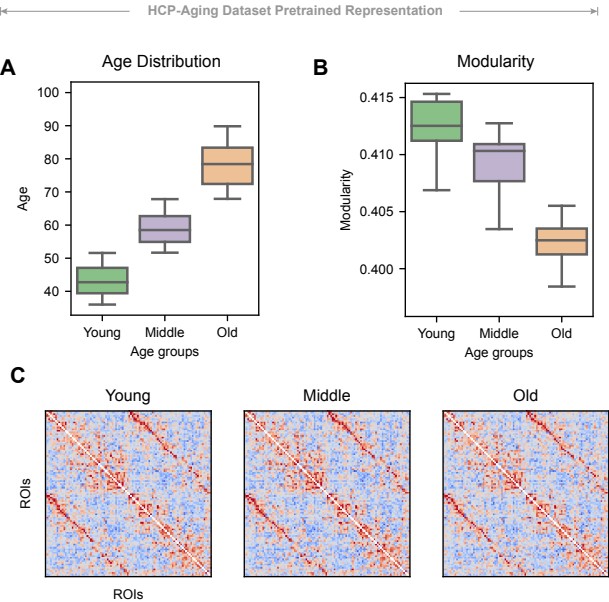

Figure 9: Age effect on learned ROI embeddings. (A) Age distribution for the three defined age groups. (B) The modularity of the ROI embedding-based functional network decreases with aging. For each group, modularity is computed 50 times using the Louvain algorithm, and the whiskers of the boxplot represent the maximum and minimum of the 50 modularity values. (C) Embedding similarity matrix of the three age groups.

The ROI embedding faithfully captures the characteristics of brain ROIs within the pre-trained fMRI dataset. To investigate the impact of aging on the acquired ROI embeddings, we partition the HCP-Aging dataset into three distinct, non-overlapping age groups (Young: 36-52, Middle: 52-68, Old: 68-100; refer to the Figure 9A for the age distribution in each group). Subsequently, we independently pre-train the SG-BrainMAE for each age group. To discern variations in ROI embeddings,

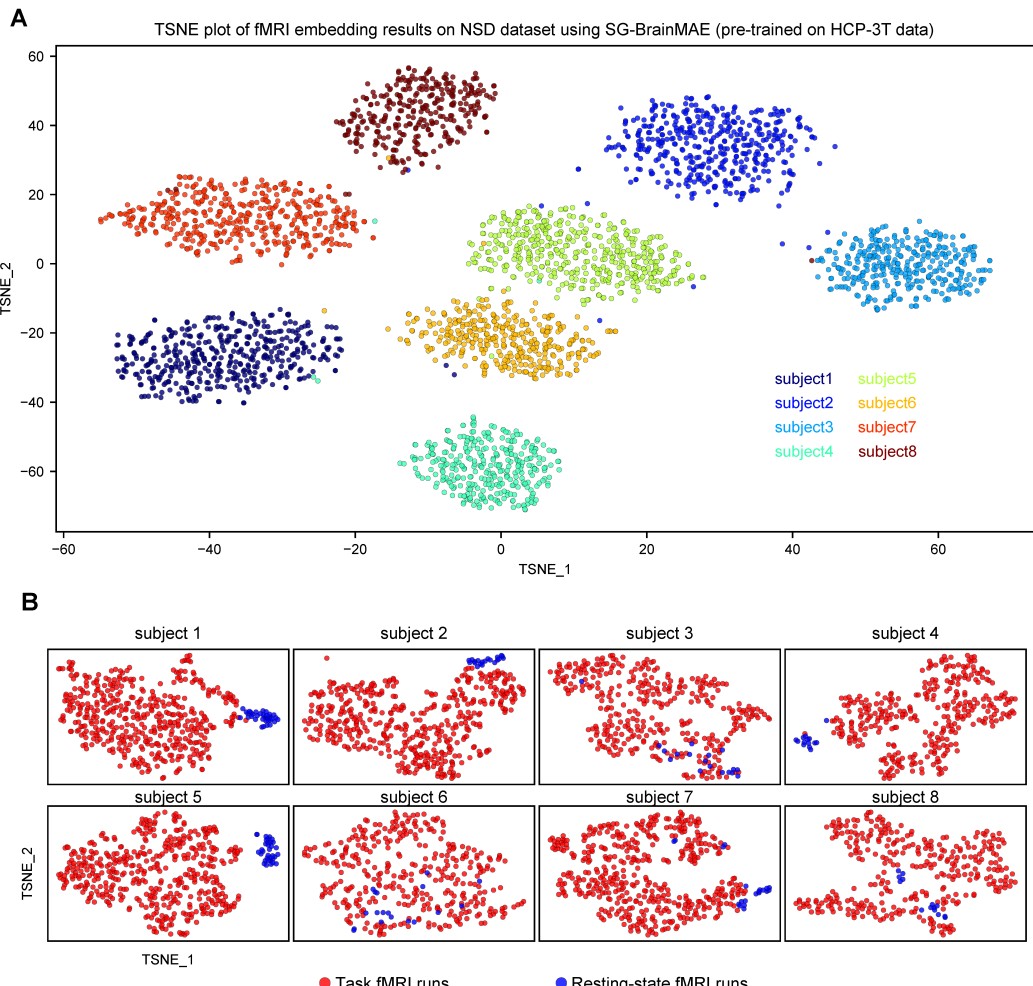

Figure 10: Representation analysis of fMRI runs using the pretrained SG-BrainMAE on the independent NSD dataset. (A) t-SNE plot of the extracted representations of fMRI runs. Each dot represents a fMRI run, and colors indicate subject identity. (B) Within-subject t-SNE plot, with red indicating task-related fMRI runs and blue representing resting-state runs.

we study the modular structure of the network constructed based on the embedding similarity matrix specific to each age group. The modularity of a network, serving as a metric in network analysis, delineates how the network can be segmented into nonoverlapping regions or the segregation of brain ROIs. The results in Figure 9B indicate a reduction in the modularity of ROI embeddings from the young age group to the old age group. This trend suggests a decline in the segregation of functional networks during aging, aligning with established findings in the neuroscience literature Sporns & Betzel (2016); Wig (2017).

## E    FMRI REPRESENTATION ANALYSIS

### E.1    SUBJECT IDENTITY

To investigate the learned representation of fMRI runs, we applied the SG-MAE pre-trained on the HCP-3T Resting-state dataset to extract fMRI representations (output of CLS token) on an independent NSD task dataset. Surprisingly, as shown in Figure 10A, we observed that the representations of fMRI runs are well separated across different subjects in the t-SNE plot. This result suggests that the pre-trained model can generalize effectively to other datasets, distinguishing individuals and encoding their traits based on their fMRI scans.

### E.2 BRAIN STATES

#### E.2.1 TASK VS. REST

Taking a step further, we aimed to analyze the representation of fMRI scans within each subject. In the NSD dataset, each subject performed multiple task sessions (image-memory task) as well as a few resting-state sessions (spontaneous rest). The Figure 10B shows the within-subject t-SNE analysis of the representation extracted by the SG-MAE pretrained with HCP3T. Remarkably, in most cases, the resting-state fMRI runs are well separated from task fMRI runs and exhibit distinct representations. This result further suggests that within individuals, the pre-trained representation carries meaningful information reflecting one's brain state.

#### E.2.2 BRAIN AROUSAL STATE

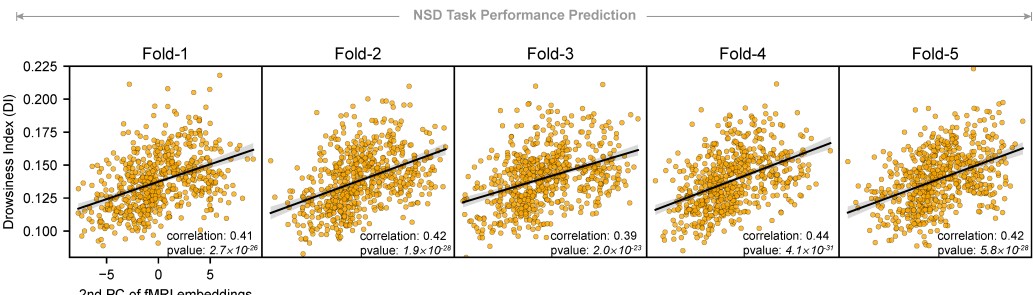

Figure 11: Strong correlation between the second principal component (PC) of the learned fMRI representation and drowsiness index (DI). Each column displays the result of each fold in the 5-fold cross-validation.

In interpreting the representations extracted by the model fine-tuned on the NSD dataset for downstream task performance prediction, we conducted PCA analysis on the fMRI representations (output of CLS token). Intriguingly, as shown in Figure 11, we observed a close relationship between the second principal component and the drowsiness index, a metric for measuring the brain arousal level Chang et al. (2016); Gu et al. (2020). This finding suggests a convergence between our data-driven approach and traditional neuroscience studies that quantify brain states with multimodal equipment. It implies that the proposed method could serve as a valuable tool to sufficiently identify brain states using fMRI alone, obviating the need for additional modalities such as EEG.

## F  HCP TRANSIENT MENTAL STATE DECODING

Table 10: Results for transient mental state decoding

| Model | Accuracy(%) | macro F1-score |
|---|---|---|
| ***Self-supervised learning-based model*** | | |
| CSM | 94.8±0.35 | 92.0 |
| Seq-BERT | 89.2±0.35 | 84.5 |
| Net-BERT | 89.8±0.48 | 85.2 |
| ***Ours*** | | |
| SG-BrainMAE | 95.4±1.2 | 94.7 |
| AG-BrainMAE | **95.5±1.5** | **94.8** |

In addition to the downstream tasks performed in section 3.3, where pretrained models are fine-tuned to predict traits or task performance from *hundreds of seconds* of fMRI signal, we here aim to further extend our study and evaluate whether BrainMAEs can effectively decode transient brain states of *tens of seconds*. We compare its performance with current state-of-the-art methods specifically designed for this purpose.

Table 11: Results of each mental state from transient mental state decoding.

| Task | Mental States | | Model F1-score(%) | |
|---|---|---|---|---|
| | Name | Duration(second) | SG-BrainMAE | AG-BrainMAE |
| Working Memory | body | 27.5 | 92.45±2.83 | 93.81±2.62 |
| | faces | 27.5 | 93.54±2.53 | 93.70±0.83 |
| | places | 27.5 | 97.18±1.37 | 97.39±1.30 |
| | tools | 27.5 | 92.75±2.84 | 92.79±2.25 |
| Gambling | win | 28.0 | 82.74±4.00 | 82.59±5.53 |
| | loss | 28.0 | 82.62±3.89 | 85.10±3.59 |
| Motor | left finger | 12.0 | 97.61±1.47 | 98.04±1.58 |
| | right finger | 12.0 | 97.65±1.15 | 97.79±1.81 |
| | left toe | 12.0 | 98.82±1.12 | 99.12±0.81 |
| | right toe | 12.0 | 97.64±2.29 | 97.62±1.87 |
| | tongue | 12.0 | 99.10±1.63 | 98.37±1.78 |
| Language | story | 25.9 | 97.87±0.86 | 97.26±1.94 |
| | math | 16.0 | 98.31±1.24 | 98.84±0.90 |
| Social | interaction | 23.0 | 98.23±0.73 | 98.20±1.21 |
| | no interaction | 23.0 | 97.98±1.51 | 97.41±1.79 |
| Relational | relational | 16.0 | 92.06±2.17 | 92.94±2.63 |
| | matching | 16.0 | 93.16±2.69 | 92.98±2.55 |
| Emotion | fear | 18.0 | 97.73±1.61 | 96.37±1.92 |
| | neutral | 18.0 | 97.60±1.70 | 97.85±0.57 |
| Rest | Rest | 864.0 | 89.11±7.02 | 87.92±5.47 |

**Dataset and Downstream Task.** Following the same experimental setup as in (Thomas et al., 2022), we use publicly available HCP-3T task datasets (Van Essen et al., 2013) and identify 20 mental states across experimental tasks. The models are fine-tuned to classify brain state from 20 mental states using fMRI signals lasting tens of seconds.

**Baselines.** We compare our methods with current state-of-the-art transformer-based self-supervised learning techniques tailored for fMRI transient state modeling. In these methods, each time point of signal is treated as a word, leveraging recent successes in NLP pre-training techniques to model fMRI transient states. Causal Sequence Modeling (CSM) is pretrained based on the principle of causal language modeling, predicting the next signal based on historical context. Sequence-BERT performs self-supervised learning by solving masked-language-modeling and next-sentence-prediction tasks. Network-BERT, a variant to Sequence-BERT, is designed to infer the entire time series of a masked network of Regions of Interest (ROIs).

**Transfer Learning.** We fine-tune the BrainMAEs similarly to other downstream tasks in section 3.3 with minor adjustments. We use BrainMAEs pretrained on the independent NSD dataset. To make the model suitable for transient state decoding, we input fixed lengths of 30 seconds of fMRI signals. For mental states with a duration of less than 30 seconds, we use the fMRI signal from the mental task block center, expanding it both left and right to achieve a total duration of 30 seconds. Resting-state samples are randomly drawn from 864-second fMRI run for a length of 30 seconds.

**Performance Comparison.** The results on the test set are shown in Table 10 and reported as mean ± std. from 5-fold cross-validation. Our proposed methods outperform other self-supervised techniques that are designed for decoding the transient fMRI signal. It's important to note that we only use 100 cortical ROIs, in contrast to baseline methods utilizing 1024 Dictionary Learning Functional Modes (DiFuMo) ROIs including subcortical areas. This comparison leads to two principal insights: firstly, cortical region activity alone might suffice for decoding mental states; secondly, given the highly correlated nature of fMRI signals, a small set of ROIs can effectively represent brain activity, potentially enabling the development of more efficient models for future research. Further-

more, AG-BrainMAE exhibits marginally enhanced performance relative to SG-BrainMAE, indicating that integrating an adaptive component is advantageous for capturing transient state changes in fMRI data. Table 11 presents detailed results for each mental state from the multi-class decoding task, demonstrating consistently high decoding accuracy that appears to be insensitive to variations in state duration.

## G   ADDITIONAL RESULTS

### G.1   ABLATION STUDY ON ROI EMBEDDING

Given that the position of the ROIs is static, there exists a possibility that the learned ROI embedding may predominantly encode information about the absolute position rather than the functional characteristics of the ROI. This hypothesis was investigated with two analyses from different perspectives.

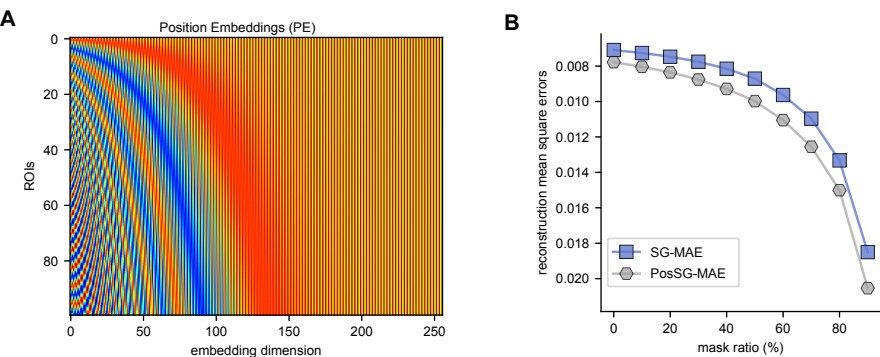

Figure 12: A. Position embedding used as ROI embeddings. B. The PosSG-BrainMAE, utilizing position embeddings, exhibited inferior reconstruction results compared to SG-BrainMAE with informative ROI embeddings.

Table 12: More Ablation analysis: HCP 3T downstream task results.

| Model | Gender | | Behaviors (measured in MAE) | | | | | | | |
|---|---|---|---|---|---|---|---|---|---|---|
| | Accuracy(%) | AUROC | PicSeq | PMT_CR | PMT_SI | PicVocab | IWRD | ListSort | LifeSatisf | PSQI |
| SG-BrainMAE | **97.49±0.15** | **97.46±0.18** | **5.06±0.21** | **1.63±0.08** | **1.24±0.04** | **3.40±0.14** | **1.11±0.04** | **4.35±0.12** | **3.64±0.27** | **1.05±0.06** |
| PosSG-BrainMAE | 95.74±1.08 | 95.98±0.96 | 5.99±0.30 | 1.83±0.09 | 1.41±0.08 | 3.80±0.08 | 1.28±0.05 | 4.83±0.20 | 4.02±0.08 | 1.22±0.08 |
| SG-BrainMAE(SL) | 96.63±1.56 | 96.62±1.55 | 5.53±0.15 | 1.72±0.06 | 1.32±0.07 | 3.61±0.10 | 1.15±0.03 | 4.64±0.06 | 3.87±0.10 | 1.14±0.04 |

Table 13: More Ablation analysis: HCP-Aging.

| Model | Gender | | Aging(MAE) |
|---|---|---|---|
| | Accuracy(%) | AUROC | |
| SG-BrainMAE | **92.67±1.07** | **92.51±1.07** | 5.75±0.44 |
| PosSG-BrainMAE | 88.38±2.93 | 88.28±3.19 | 6.66±0.71 |
| SG-BrainMAE(SL) | 91.29±1.34 | 91.45±1.33 | **5.68±0.31** |

Table 14: More Ablation analysis: NSD.

| Model | Task Accuracy | RT(ms) |
|---|---|---|
| SG-BrainMAE | **0.069±0.004** | **90.678±1.767** |
| PosSG-BrainMAE | 0.080±0.004 | 100.064±4.439 |
| SG-BrainMAE(SL) | 0.078±0.003 | 98.469±1.675 |

**Analysis 1.** We substitute ROI embedding in the SG-BrainMAE model with position embedding, which is then frozen throughout the pretraining phase on the HCP-3T dataset while keeping other model components unchanged. This adapted model is named PosSG-BrainMAE. Evaluations involving both the reconstruction of masked signals on an independent HCP-7T dataset (see Figure 12) and performance in downstream tasks (see Tabel 12, 13, and 14) reveal a decreased efficacy compared to SG-BrainMAE. This decline in performance justifies the valuable ROI information contained in learned ROI embeddings, extending beyond mere positional information.

**Analysis 2.** We cyclic shift each ROI's fMRI signal by random time steps. By doing this, the shifted fMRI signal exihibit two properties: 1. each fMRI signal itself is merely changed; 2. elimination of

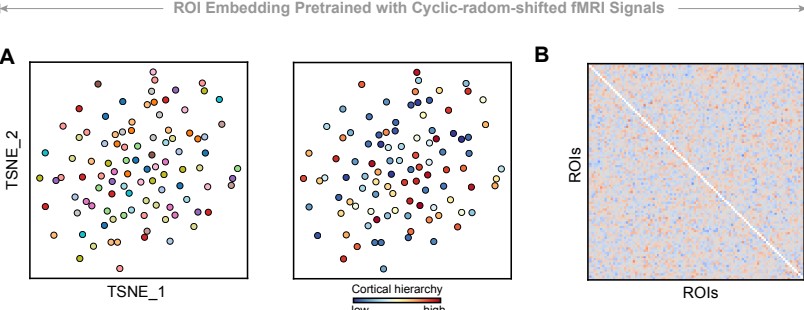

Figure 13: Ablation analysis 2: evaluation of pre-trained ROI embeddings with cyclic-random-shifted fMRI signals. (A) A t-SNE plot of ROI embeddings with ROI colored based on Yeo-17 network (Left) and cortical hierarchy (Right). (B) ROI embedding similarity matrix with indiscernible inter-network relationship.

the inter-relationship between pairs of ROIs. We then follow the same pre-training procedure using this modified dataset. The ROI embeddings learned from this dataset did not exhibit functional specificity and inter-regional connectivity (see Figure 3), in contrast to those learned from the actual dataset (see Figure 13). These findings provide additional evidence that the proposed method learns meaningful ROI information from the dataset, including its relationship to other ROIs.

### G.2 ABLATION STUDY ON SELF-LOOP REMOVAL

Removing self-loops in our model avoids the attention mechanism favoring its own node, encouraging it to aggregate signals from other relevant nodes. This approach is akin to a group voting system and helps reduce sensitivity to input noise. To validate this design choice, we conducted a comparative analysis for downstream tasks between SG-BrainMAE and a similar model that includes self-loops, named SG-BrainMAE(SL). The results, presented in Tables 12, 13, and 14, show a slight decrease in performance for SG-BrainMAE(SL), indicating the effectiveness of excluding self-loops in our model.

### G.3 COMPARISON WITH SELF-SUPERVISED LEARNING ON GENDER CLASSIFICATION ON HCP-3T DATASET

Table 15: HCP, Gender Prediction

| Model | Accuracy(%) |
|---|---|
| TFF | 94.09 |
| *Ours* | |
| **SG-BrainMAE** | **97.49±0.15** |
| **AG-BrainMAE** | **97.13±0.56** |

We conducted comparative analysis between our method and another recent self-supervised learning approach, named TFF, which employs 3D Convolutional Neural Networks (CNNs) and transformer to extract volumetric representations of fMRI data at each time point, with pre-training via auto-encoding. The results, as shown in Table 15, demonstrate that our model outperforms TFF in the HCP gender prediction downstream task.

### G.4 COMPARISON WITH TRADITIONAL MACHINE LEARNING MODELS

Given the effectiveness and prevalence of traditional machine learning (ML) models in neuroimaging communities, this section focuses on assessing the added performance benefits of utilizing complex deep learning-based methods in comparison to these simpler ML models.

For regression tasks, we consider a suite of linear models, including ordinary linear regression, ridge regression, and elastic net. In the context of classification tasks, we explore the use of logistic regression, linear Support Vector Machine (SVM), and Random Forest models. Each of these models is trained to make predictions based on the flattened upper-triangle of the Functional Connectivity (FC) connectivity matrix.

We employ cross-validated grid search approach, with specific ranges and increments for each model for hyperparameter selection of each ML model:

- **Support Vector Machine (SVM)**: We vary the L2 regularization coefficient from 0.1 to 10, with an increment of 0.5.
- **Logistic Regression**: The L2 regularization coefficient is tuned from 0.1 to 10, with an increment of 0.5.
- **Random Forest**: Three key parameters are tuned: a. Number of trees, ranging from 1 to 250 with increments of 50. b. Maximum depth of each tree, from 5 to 50 with increments of 10. c. Minimum samples required to split a node, from 5 to 100 with increments of 20.
- **Ordinary Linear Regression**: This model did not require hyperparameter tuning.
- **Ridge Regression**: The L2 regularization coefficient is tuned from 0 to 10, with an increment of 0.5.
- **Elastic Net Regression**: a. The coefficient of the L2 penalty (ridge regression component) is tuned from 0 to 10, with increments of 0.5. b. The coefficient of the L1 penalty (lasso regression component) is adjusted from 0 to 1, with increments of 0.2.

For models requiring multiple hyperparameters, we train for each possible combination. The best-performing model is selected based on its performance on the validation set, using Mean Squared Error (MSE) for regression tasks and accuracy for classification tasks.

The results of this comparative analysis across three different downstream tasks are shown in Table 16, 17, 18 and 19. It reveals that for classification tasks, traditional ML methods demonstrate performance levels comparable to those of baseline deep learning methods. This observation can be attributed to the well-established understanding that the functional connectivity matrix harbors significant information pertinent to human traits and age. However, for more complex regression tasks, such as task performance prediction, which necessitate inferring intricate brain states from the dynamics of fMRI signals, ML models often exhibit less satisfactory performance. In such scenarios, deep learning methods, endowed with their robust capability for representation learning, are able to achieve markedly superior results.

Table 16: HCP 3T Dataset (FIX-ICA), Gender Classification

| Model | Gender | | | | |
|---|---|---|---|---|---|
| | Accuracy(%) | AUROC | specificity(%) | sensitivity(%) | F1 Score(%) |
| *FIX-FC* | | | | | |
| BrainNetTF | 94.11±0.98 | 94.39±0.90 | 95.36±0.70 | 93.24±2.08 | 93.69±1.01 |
| VanillaTF | 90.00±1.05 | 89.93±0.94 | 91.36±1.25 | 88.32±2.51 | 88.71±1.10 |
| BrainNetCNN | 90.68±1.80 | 90.89±1.55 | 93.82±1.64 | 87.40±3.80 | 89.56±1.44 |
| *Dynamic-FC* | | | | | |
| STAGIN-SERO | 88.73±1.36 | 88.69±1.41 | 89.99±1.71 | 86.29±1.74 | 86.80±0.90 |
| STAGIN-GARO | 88.34±0.94 | 88.33±0.91 | 89.50±2.90 | 86.76±4.73 | 97.18±0.13 |
| FBNETGNN | 88.05±0.15 | 87.93±0.97 | 89.60±1.21 | 86.11±1.32 | 86.51±0.89 |
| *Machine Learning Model* | | | | | |
| SVM | 87.55±1.79 | 87.53±1.91 | 90.16±2.31 | 84.31±2.05 | 85.76±2.07 |
| Logistic Regression | 92.16±0.77 | 92.10±0.71 | 93.18±1.17 | 90.91±2.04 | 91.16±0.84 |
| Forest | 77.16±2.63 | 77.42±2.64 | 85.23±2.05 | 67.09±4.21 | 72.29±3.43 |
| *Ours* | | | | | |
| **SG-BrainMAE** | **97.49±0.15** | **97.46±0.18** | **97.66±0.91** | **97.28±1.19** | **97.18±0.13** |
| **AG-BrainMAE** | **97.13±0.56** | **97.17±0.61** | **97.62±0.95** | **96.14±0.76** | **96.76±0.61** |

Table 17: HCP 3T Dataset (FIX-ICA), Behavior Regression

| Model | Behaviors (measured in MAE) | | | | | | | |
|---|---|---|---|---|---|---|---|---|
| | PicSeq | PMT_CR | PMT_SI | PicVocab | IWRD | ListSort | LifeSatisf | PSQI |
| *FIX-FC* | | | | | | | | |
| BrainNetTF | 7.11±0.22 | 2.28±0.11 | 1.72±0.12 | 4.70±0.10 | 1.56±0.03 | 5.96±0.07 | 4.96±0.22 | 1.49±0.05 |
| VanillaTF | 8.19±0.46 | 2.73±0.07 | 2.13±0.07 | 5.93±0.14 | 1.81±0.08 | 6.91±0.21 | 5.71±0.08 | 1.68±0.05 |
| BrainNetCNN | 10.21±0.22 | 3.25±0.13 | 2.64±0.14 | 6.65±0.27 | 2.26±0.05 | 8.51±0.20 | 7.12±0.24 | 1.68±0.05 |
| *Dynamic-FC* | | | | | | | | |
| STAGIN-SERO | 10.22±0.15 | 3.49±0.05 | 2.70±0.06 | 6.78±0.20 | 2.26±0.05 | 8.51±0.20 | 7.12±0.24 | 2.12±0.04 |
| STAGIN-GARO | 10.26±0.18 | 3.44±0.10 | 2.69±0.09 | 6.92±0.30 | 2.25±0.04 | 8.52±0.26 | 7.09±0.35 | 2.08±0.04 |
| FBNETGNN | 8.62±0.21 | 2.93±0.11 | 2.34±0.11 | 5.83±0.15 | 1.96±0.04 | 7.31±0.10 | 6.09±0.10 | 1.81±0.03 |
| *ML Model* | | | | | | | | |
| Oridinary Regression | 11.23±0.25 | 4.05±0.07 | 3.34±0.07 | 7.50±0.13 | 2.58±0.08 | 9.90±0.26 | 8.05±0.18 | 2.50±0.07 |
| Ridge | 8.91±0.23 | 3.18±0.10 | 2.63±0.10 | 6.05±0.10 | 2.05±0.06 | 7.70±0.23 | 6.42±0.02 | 2.01±0.09 |
| ElasticNet | 10.83±0.14 | 4.01±0.03 | 3.29±0.04 | 7.44±0.22 | 2.35±0.05 | 9.21±0.19 | 7.29±0.28 | 2.15±0.07 |
| *Ours* | | | | | | | | |
| **SG-BrainMAE** | **5.06±0.21** | **1.63±0.08** | **1.24±0.04** | **3.40±0.14** | **1.11±0.04** | **4.35±0.12** | **3.64±0.27** | **1.05±0.06** |
| **AG-BrainMAE** | **5.09±0.05** | **1.67±0.10** | **1.28±0.06** | **3.34±0.11** | **1.13±0.03** | **4.37±0.06** | **3.58±0.17** | **1.07±0.05** |

Table 18: HCP Aging

| Model | Gender | | | | | Aging(MAE) |
|---|---|---|---|---|---|---|
| | Accuracy(%) | AUROC(%) | specificity(%) | sensitivity(%) | F1 Score(%) | |
| *FIX-FC* | | | | | | |
| BrainNetTF | 90.21±3.81 | 90.73±2.85 | 87.97±9.26 | 91.84±6.56 | 89.35±3.49 | 6.15±0.71 |
| VanillaTF | 88.96±2.16 | 88.76±2.21 | 89.22±2.30 | 88.56±3.01 | 87.54±2.53 | 6.78±0.56 |
| BrainNetCNN | 88.83±1.52 | 88.74±1.58 | 89.87±2.85 | 87.92±3.18 | 87.48±1.20 | 8.71±0.62 |
| *Dynamic-FC* | | | | | | |
| STAGIN-SERO | 82.37±1.66 | 82.57±1.36 | 85.23±4.68 | 76.00±6.51 | 77.91±3.14 | 8.96±0.47 |
| STAGIN-GARO | 80.67±0.81 | 80.58±1.03 | 84.63±4.05 | 77.95±4.43 | 78.81±3.27 | 8.65±0.28 |
| FBNETGNN | 89.50±3.58 | 89.34±3.49 | 90.04±1.94 | 89.05±5.18 | 88.35±3.45 | 6.68±1.00 |
| *ML Model* | | | | | | |
| SVM | 86.04±0.97 | 86.23±0.98 | 90.87±0.66 | 79.84±1.76 | 83.36±1.48 | - |
| Logistic Regression | 89.49±1.27 | 89.41±1.23 | 90.92±0.81 | 87.66±2.17 | 88.20±1.50 | - |
| Forest | 73.75±1.44 | 74.42±0.94 | 85.82±1.46 | 58.35±4.11 | 66.05±2.47 | - |
| OrdinaryRegression | - | - | - | - | - | 7.63±0.21 |
| Ridge | - | - | - | - | - | 7.08±0.20 |
| ElasticNet | - | - | - | - | - | 9.43±0.61 |
| *Ours* | | | | | | |
| **SG-BrainMAE** | **92.67±1.07** | **92.51±1.07** | **97.66±0.91** | **97.28±1.19** | **97.18±0.13** | **5.78±0.44** |
| **AG-BrainMAE** | **91.12±1.99** | **91.15±2.03** | **97.62±0.95** | **96.14±0.76** | **96.76±0.61** | **6.49±1.00** |

Table 19: Task Performance Prediction on NSD dataset

| Model | Task Accuracy | RT(ms) |
|---|---|---|
| *FIX-FC* | | |
| BrainNetTF | 0.070±0.003 | 92.344±2.343 |
| VanillaTF | 0.075±0.004 | 96.252±2.133 |
| BrainNetCNN | 0.078±0.004 | 102.911±2.225 |
| *Dynamic-FC* | | |
| STAGIN-SERO | 0.089±0.003 | 116.635±2.197 |
| STAGIN-GARO | 0.091±0.002 | 116.130±2.099 |
| FBNETGNN | 0.074±0.005 | 95.349±2.320 |
| *ML Model* | | |
| Ordinary Regression | 0.117±0.003 | 157.460±3.788 |
| Ridge | 0.093±0.003 | 129.609±2.518 |
| ElasticNet | 0.110±0.004 | 174.946±2.585 |
| *Ours* | | |
| **SG-BrainMAE** | **0.069±0.004** | **90.678±1.767** |
| **AG-BrainMAE** | **0.070±0.004** | **92.154±2.265** |

Table 20: Result for behavior prediction on minimal preprocessed fMRI using model finetuned with ICA-FIX pipleline

| Model | Gender | | Behaviors (measured in MAE) | | | | | | | |
| | Accuracy(%) | AUROC | PicSeq | PMT_CR | PMT_SI | PicVocab | IWRD | ListSort | LifeSatisf | PSQI |
|---|---|---|---|---|---|---|---|---|---|---|
| *FIX-FC* | | | | | | | | | | |
| BrainNetTF-OCR | 79.62±1.57 | 80.15±0.65 | 9.76±0.38 | 3.52±0.08 | 2.76±0.04 | 6.64±0.18 | 2.15±0.05 | 8.21±0.11 | 6.83±0.20 | 1.98±0.05 |
| BrainNetTF-Vanilla | 75.53±1.58 | 75.41±1.51 | 10.56±0.36 | 3.67±0.03 | 2.93±0.04 | 7.54±0.46 | 2.35±0.06 | 8.90±0.15 | 7.47±0.17 | 2.12±0.07 |
| BrainNetCNN | 73.34±3.37 | 74.85±1.8 | 11.04±0.13 | 4.10±0.21 | 3.42±0.24 | 7.65±0.24 | 2.43±0.08 | 9.46±0.08 | 7.37±0.30 | 2.17±0.07 |
| *Dynamic-FC* | | | | | | | | | | |
| STAGIN-SERO | 69.22±3.56 | 69.98±2.86 | 11.19±0.35 | 3.96±0.15 | 3.11±0.14 | 7.93±0.38 | 2.15±0.05 | 8.21±0.11 | 6.83±0.20 | 1.98±0.05 |
| STAGIN-GARO | 70.03±1.86 | 71.11±1.26 | 11.67±0.21 | 3.91±0.13 | 3.07±0.16 | 8.01±0.31 | 2.42±0.09 | 9.60±0.35 | 7.56±0.15 | 2.25±0.05 |
| FBNETGNN | 72.10±1.80 | 72.19±1.87 | 11.41±0.30 | 3.99±0.25 | 3.21±0.22 | 8.02±0.44 | 2.60±0.06 | 10.09±0.30 | 8.03±0.38 | 2.36±0.09 |
| *Ours* | | | | | | | | | | |
| **SG-BrainMAE** | **86.05±2.71** | **86.01±2.73** | **8.76±0.19** | **3.11±0.07** | **2.41±0.05** | **5.83±0.24** | **1.92±0.04** | **7.46±0.2.** | **6.07±0.22** | **1.76±0.09** |
| **AG-BrainMAE** | **83.93±1.57** | **84.06±1.29** | **8.89±0.23** | **3.15±0.06** | **2.46±0.08** | **5.89±0.19** | **1.94±0.05** | **7.48±0.17** | **6.11±0.24** | **1.80±0.07** |

Table 21: Ablation Study on HCP 3T behavior prediction

| Model | Gender | | Behaviors (measured in MAE) | | | | | | | |
| | Accuracy(%) | AUROC | PicSeq | PMT_CR | PMT_SI | PicVocab | IWRD | ListSort | LifeSatisf | PSQI |
|---|---|---|---|---|---|---|---|---|---|---|
| SG-BrainMAE | **97.49±0.15** | **97.46±0.18** | 5.06±0.21 | 1.63±0.08 | **1.24±0.04** | 3.40±0.14 | 1.11±0.04 | 4.35±0.12 | 3.64±0.27 | 1.05±0.06 |
| AG-BrainMAE | 97.13±0.56 | 97.17±0.61 | 5.07±0.09 | 1.67±0.10 | 1.28±0.06 | 3.34±0.11 | 1.13±0.03 | 4.37±0.06 | 3.58±0.17 | 1.07±0.05 |
| vanilla-BrainMAE | 96.66±0.94 | 96.85±0.81 | **4.85±0.16** | **1.61±0.07** | 1.25±0.07 | **3.23±0.08** | **1.08±0.05** | **4.12±0.15** | **3.30±0.16** | **0.99±0.04** |
| vanilla-BrainAE | 94.11±1.02 | 94.07±1.09 | 7.63±0.27 | 2.50±0.12 | 1.92±0.11 | 5.01±0.17 | 1.67±0.04 | 6.45±0.28 | 5.40±0.13 | 1.59±0.05 |

Table 22: The distinction between different model variants

| Model variants | Transient state encoder (TSE) | Pretraining |
|---|---|---|
| *Primary* | | |
| SG-BrainMAE | Three ROI embedding informed attention block | Masked-Autoencoding |
| AG-BrainMAE | Two self-attention blocks stacked on top of an ROI embedding informed attention block | Masked-Autoencoding |
| *Other* | | |
| vanilla-BrainMAE | Three self-attention blocks | Masked-Autoencoding |
| vanilla-BrainAE | Three self-attention blocks | Autoencoding |
| PosSG-BrainMAE | Three absolute position embedding informed attention block | Masked-Autoencoding |
| SG-BrainMAE (SL) | Three ROI embedding informed attention block where attention was computed without self-loop removal | Masked-Autoencoding |

## H  ADDITIONAL RECONSTRUCTION RESULTS

Figure 14: Example reconstruction results on HCP-7T using HCP-3T pretrained model.

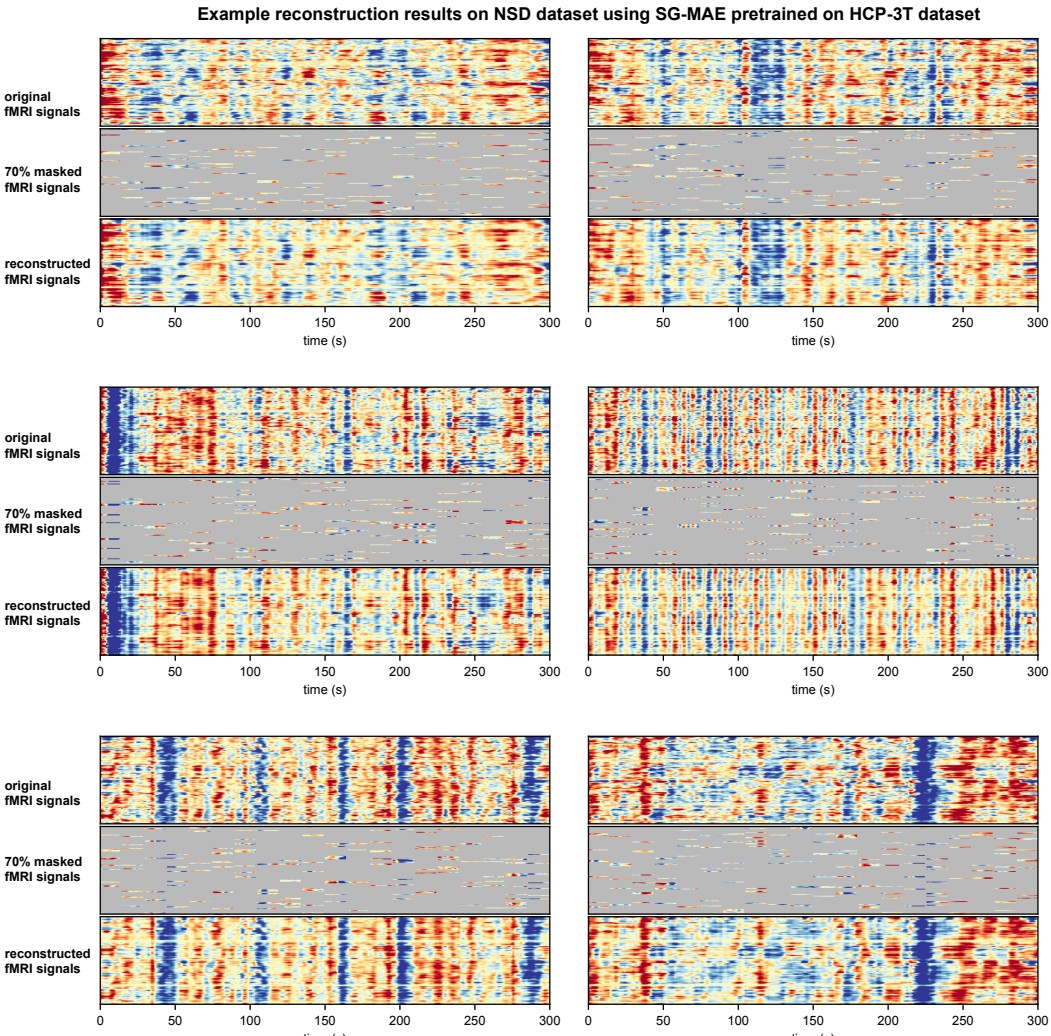

Figure 15: Example reconstruction results on NSD using HCP-3T pretrained model.

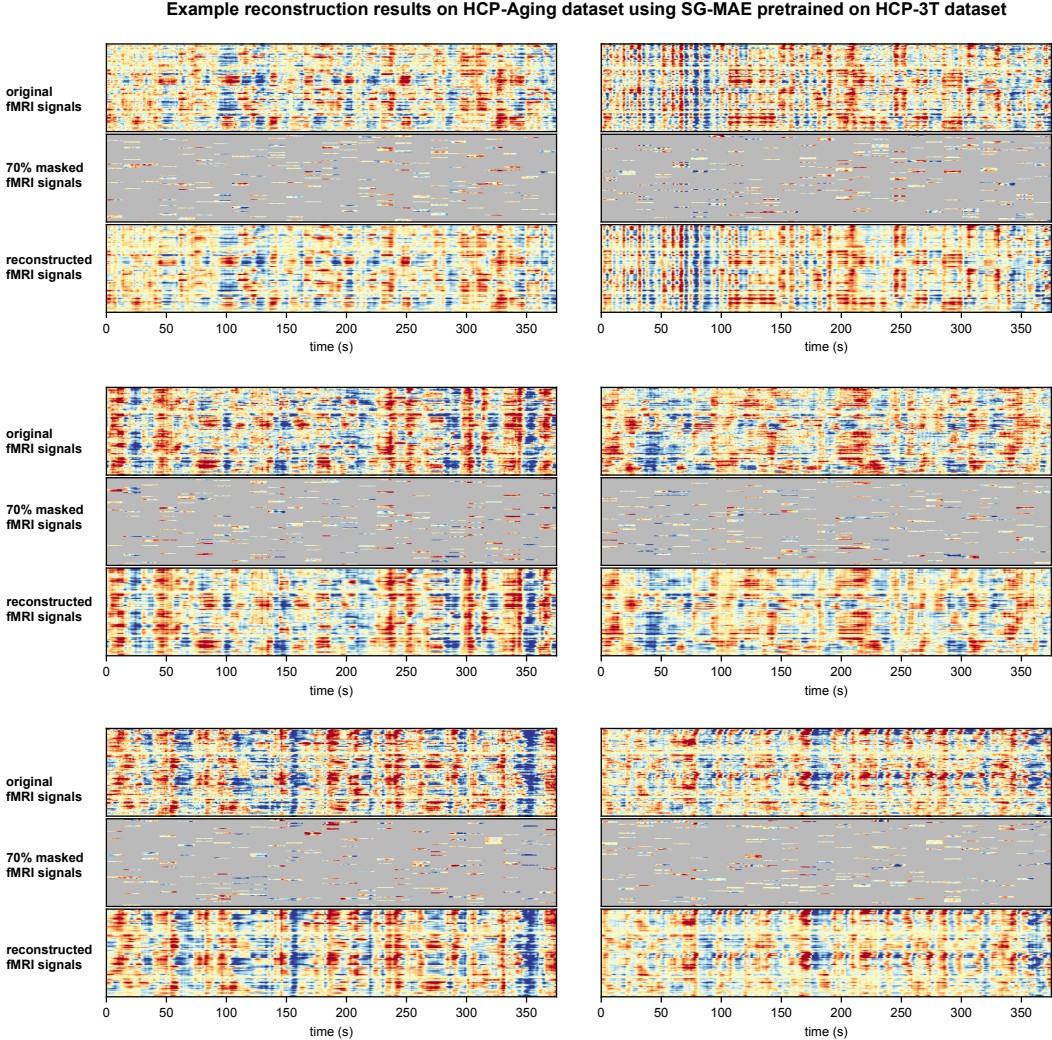

Figure 16: Example reconstruction results on HCP-Aging dataset using HCP-3T pretrained model.

