# OpenReview forum: "Learning transferrable and interpretable representation for brain network"
_ICLR.cc/2024/Conference — Submitted to ICLR 2024_

### Official Review · Reviewer_nn5J · 2023-10-25

**Soundness:** 3 good
**Presentation:** 3 good
**Contribution:** 2 fair
**Rating:** 5
**Confidence:** 5

**Summary:**

This paper proposes a Brain Masked Auto-Encoder (BrainMAE) that consists of two main components: an embedding-based graph attention mechanism and a self-supervised masked auto-encoding framework. BrainMAE uses a static graph transient state encoder (SG-TSE) and dynamic graph TSE (DG-TSE) to learn ROI representations. The trained ROI embeddings showed the distinctive traits of ROIs, resulting in the consistency of ROI embeddings across the different datasets. BrainMAE achieved improved performances in three distinct downstream tasks.

**Strengths:**

The paper is written clearly and well-organized.
The analysis of ROI embeddings for inter-individual consistency demonstrates the generalizability of ROI embeddings.
The paper conducts numerous experiments to validate the efficacy of the proposed BrainMAE, outperforming the comparative methods.

**Weaknesses:**

The proposed BrainMAE is based on self-supervised masked auto-encoding, so it needs to be compared with the existing self-supervised learning-based models, including the following:
[1] Shi, Chenwei, et al. "Self-supervised pretraining improves the performance of classification of task functional magnetic resonance imaging." Frontiers in Neuroscience 17 (2023).
[2] Malkiel, Itzik, et al. "Self-supervised transformers for fMRI representation." International Conference on Medical Imaging with Deep Learning. PMLR, 2022.
[3] Thomas, Armin, Christopher Ré, and Russell Poldrack. "Self-supervised learning of brain dynamics from broad neuroimaging data." Advances in Neural Information Processing Systems 35 (2022): 21255-21269.

Many fMRI studies have demonstrated that the length of each time segment significantly affects the performance. Most related studies have empirically converged to window size values between 30 and 60 seconds [4]. However, the proposed method uses 15 seconds, which is too short for the window size. Do you have a rationale for this window size?
[4] Savva, Antonis D., Georgios D. Mitsis, and George K. Matsopoulos. "Assessment of dynamic functional connectivity in resting‐state fMRI using the sliding window technique." Brain and Behavior 9.4 (2019): e01255.

Since the word embedding includes information about its meaning, the word embeddings in NLP can be used in all sentence positions.
However, the position of ROIs is never changed in training sessions, which could limit the model's ability to learn ROI traits but could learn only absolute position information. It can be considered as learnable positional encoding. Since the graph attention mechanism is permutation-invariant, the changes in ROI order should not affect the performance if the ROI embeddings have their characteristics. Even though the paper shows the evaluation of pretrained ROI embeddings in Figure 3, it still needs additional evidence that these embeddings do not incorporate absolute position information.

After learning, the fixed ROI embeddings E are used repeatedly in SG-TSE for an attention mechanism. However, since the same attention is obtained with a fixed Q and K, there is no need to use a transformer block.

One of the main reasons for learning and exploiting the ROI embeddings was to mitigate and circumvent the inherent noise in the fMRI signal. However, from the experimental results, the authors concluded that the lower performance of DG-BrainMAE than that of SG-BrainMAE was due to such inherent noise. These are conflicting.

**Questions:**

Check the comments in Weaknesses.

---

> ### Author Response · Authors · 2023-11-18
> **Response to Reviewer nn5J (1/2)**
>
> Thank you for this insightful review! We sincerely appreciate your time in reading the paper. The comments and concerns are extremely helpful for us to improve the paper.  We've incorporated several new analyses to address these concerns.
>
> > Q1: Comparison with the existing self-supervised learning-based models.
>
> Thank you for raising this concern. Following your suggestion, we added a downstream experiment in $\textbf{Appendix F}$ of our revised submission using the HCP Task Dataset, same to the setup in [1]. The models are trained to make classification between 20 mental states using fMRI signals, with each state lasting tens of seconds — a notable difference from the downstream tasks in section 3.3. Despite this, $\textbf{Tables 10 and 11}$ show our models outperforming other state-of-the-art, self-supervised learning models designed for transient state modeling. Further details are available in $\textbf{Appendix F}$.
>
> Additionally, we added $\textbf{Appendix G.3}$ to compare our model's performance in gender prediction with [2], as shown in $\textbf{Table 15}$. Our model outperforms TTF by a significant margin.
>
> We haven't included a comparison with [3] due to the unavailability of their code. However, we would be happy to compare with them and following their exact experiment setup for our model evaluation if you have any additional concerns.
>
> > Q2: Why use 15 seconds for the length of time segment?
>
> Thanks for bringing up this question. It is true that computing dynamic FC usually requires longer time window (30 to 60 seconds) to handle spurious connectivity as well as intrinsic noise within the fMRI. However, distinct from computing dynamic FC, our model exhibits another advancement by embracing recent insights into "transient state" in neuroscience research. The slow cortical fluctuations are hypothesized to fluctuate at ~0.1 Hz [4], and in [5] a 10s time window was used for track the state changes. On the other hand, recent study suggest brain exhibits spatial-temporal dynamics has been shown as a major contributor to functional connectivity (FC). These fluctuations take the form from 10 seconds to 20 seconds [6][7]. Therefore, in our study, to incorporate such new understanding about the brain, we choose the 15 seconds as the length of the time segments. For handling the fMRI intrinsic noise, we propose the embedding-informed attention module with constraints encoded within the ROI embeddings.
>
> > Q3: Graph attention mechanism is permutation-invariant.
>
> Apologies for the confusion. To clarify, in our implementation of embedding-informed graph attention, we use a transformer block, where the ROI embeddings are linearly transformed using different learnable weight matrices for the key and query. This design makes the adjacency matrix inherently asymmetric, enabling it to capture the asymmetric information flow between ROIs. The introduction of these learnable weight matrices means that the attention mechanism $\textbf{isn't permutation invariant}$. We have thoroughly revised Section 2.1 in our revised submission to better explain this aspect.
>
> > Q4: More evidence to justify the learned ROI embeddings does not only incorporate absolute position information.
>
> Thanks for bringing up the insightful comments on the ROI embedding. To some extent, we believe the ROI embedding learn the position information as it itself has real positional meaning in the brain. But other than that, the ROI embeddings are designed to learn the functional information of the ROIs. To justify this hypothesis, other than the results shown in Figure 3, we additionally performed two analyses from different perspectives.
>
> 1. Age Effect on Learned ROI Embeddings ($\textbf{Appendix D.3}$): We partition the HCP-Aging dataset into young, middle, old age groups and learn the ROI embeddings for each of the groups. We form graphs based on the similarity of the ROI embeddings and study the modualr structure of the resulting graph. The resulting modularity shows decrease with age, consistent with established findings in the neuroscience literature [8]. This result suggests that indeed there is functional information contained within the ROI embeddings that reflect aging.
>
> 2. Replacing ROI Embeddings with Position Embedding ($\textbf{Appendix G.1 Analysis1}$): We replace the learnable ROI embedding with fixed position embedding, and follow the same pre-training and fine-tuning procedure. The resulting model's performance significantly dropped in downstream tasks compared to SG-BrainMAE, highlighting the importance of learned ROI information.
>
> In summary, the ROI embeddings contain valuable information, evident in age-related differences and enhanced transfer learning performance.

---

> > ### Author Response · Authors · 2023-11-18
> > **Response to Reviewer nn5J (2/2)**
> >
> > > Q5: There is no need to use a transformer block as same attention is repeatedly derived from fixed learned ROI embeddings.
> >
> > Apologies for the confusion. To clarify, in our ROI embedding-informed attention module, the ROI embeddings undergo two separate learnable linear transformations before the attention computation.
> > $$
> > A_{ij} = s(W_qe_i, W_ke_j).
> > $$
> > where, $s: R^d\times R^d \longrightarrow R$ is the similarity measurement of two vectors, e.g., scaled dot-product and $e_i$, $e_j$ are the embedding vectors for node $i$ and $j$ respectively. The weight matrices $W_q, W_k\in R^{d\times d}$ are learnable and introduce asymmetry into $A$.
> >
> > This design reflects the asymmetric nature of functional relationships between ROIs in the brain. Therefore, this is the same as the attention performed in the transformer block when we replace the query and key with the ROI embeddings.
> >
> > > Q6: One of the main reasons for learning and exploiting the ROI embeddings was to mitigate and circumvent the inherent noise in the fMRI signal. However, from the experimental results, the authors concluded that the lower performance of DG-BrainMAE than that of SG-BrainMAE was due to such inherent noise. These are conflicting.
> >
> > 1. We politely disagree "the author concluded that the lower performance of DG-BrainMAE than that of SG-BrainMAE was due to such inherent noise" because we do not make such conclusion. This misunderstanding might stem from section 3.3.3 discussing the performance of baseline dynamic FC models versus fixed FC models.
> >
> > 2. Our ablation study in section 3.3.4 demonstrates the effectiveness of ROI embedding-informed attention. Specifically, we found that DG-BrainMAE outperforms FDG-BrainMAE in the downstream tasks.
> >
> > 3. The DG-BrainMAE, introduced as a complement to SG-BrainMAE, focuses on the dynamic reconfiguration of ROI connectivity. Our experiments show that DG-BrainMAE excels SG-BrainMAE in identifying transient mental state fluctuations (Appendix F of our revised submission).
> >
> > ### References
> > [1] A. Thomas, C. R ́e, and R. Poldrack. Self-supervised learning of brain dynamics from broad neuroimaging data. Advances in Neural Information Processing Systems, 35:21255–21269, 2022
> >
> > [2] I. Malkiel, G. Rosenman, L. Wolf, and T. Hendler. Self-supervised transformers for fmri representation. In International Conference on Medical Imaging with Deep Learning, pages 895–913. PMLR, 2022
> >
> > [3] C. Shi, Y. Wang, Y. Wu, S. Chen, R. Hu, M. Zhang, B. Qiu, and X. Wang. Self-supervised pretraining improves the performance of classification of task functional magnetic resonance imaging. Frontiers in Neuroscience, 17, 2023.
> >
> > [4] K. Shen, R. M. Hutchison, G. Bezgin, S. Everling, and A. R. McIntosh. Network structure shapes spontaneous functional connectivity dynamics. Journal of Neuroscience, 35(14):5579–5588, 2015.
> >
> > [5] J. M. Shine, P. G. Bissett, P. T. Bell, O. Koyejo, J. H. Balsters, K. J. Gorgolewski, C. A. Moodie, and R. A. Poldrack. The dynamics of functional brain networks: integrated network states during cognitive task performance. Neuron, 92(2):544–554, 2016.
> >
> > [6] T. Bolt, J. S. Nomi, D. Bzdok, J. A. Salas, C. Chang, B. Thomas Yeo, L. Q. Uddin, and S. D. Keilholz. A parsimonious description of global functional brain organization in three spatiotemporal patterns. Nature neuroscience, 25(8):1093–1103, 2022.
> >
> > [7] B. Yousefi and S. Keilholz. Propagating patterns of intrinsic activity along macroscale gradients coordinate functional connections across the whole brain. NeuroImage, 231:117827, 2021.
> >
> > [8] O. Sporns and R. F. Betzel. Modular brain networks. Annual review of psychology, 67:613–640, 2016.

---

### Official Review · Reviewer_DteR · 2023-10-26

**Soundness:** 2 fair
**Presentation:** 3 good
**Contribution:** 3 good
**Rating:** 6
**Confidence:** 4

**Summary:**

The paper explores the use of functional magnetic resonance imaging (fMRI) and its representation as graph of regions of interest (ROIs) to model the human brain. It categorises these graphs based on the way the functional connectivity (FC) is handled: (1) Fixed-FC, which relies on the FC representing the linear temporal correlations within the brain network, and (2) Dynamic-FC  which aims to model the evolving FC profile over time. Each one of these two approaches has their own pros and cons in the literature. To address this problem, the paper introduces the Brain Masked Auto-Encoder (BrainMAE), which consistently outperforms existing models in various tasks. Furthermore, leveraging the model's inherent interpretability, the paper provides insights into how the model its making its decisions. BrainMAE combines a graph attention mechanism and masked autoencoding to effectively capture the dynamics of brain activity while handling the inherent noise in fMRI data.

**Strengths:**

I found the approach in the paper interesting and novel, with the key strength being the obvious consistent outperformance to other models in the literature. Differently to common papers in this area, the paper systematically analysed both classification and regression tasks, the latter being known to be more challenging and thus sometimes not explored in methodological literature. With the exceptions that I explain in "Weaknesses", overall the paper is clear and the reader can understand what are the different components, as well as their distinct contributions in the ablation analysis provided. The way the paper introduced the three characteristics that each ROI embedding should have (in section 2.1) is also a strength of this paper, and thus making the evaluation of the paper significant, as it is later explored in a satisfactory way in section 3.2.3. The interpretability part is insightful and positive - while showing that expected results are achieved (e.g., with gender being in the 1st principal component), it also shows other unexpected results for future discussion by the community.

I would like to say that the apparent similarities between the representations of brain regions and words in natural languages (as described in "ROI Embeddings" in Section 2.1) is interesting and new to me, and therefore I found this framing a (weaker) strength of the paper. Finally, the balance between figures and tables is devised well, bringing more points to both the quality and clarity of this work.

Based on the good results compared with previous literature, and the novel way to combine previous methods in deep learning literature into a new field, I recommend acceptance. I only recommend marginally above the acceptance threshold at the moment because of what I've written in the Weaknesses and Questions sections of my review.

**Weaknesses:**

Beyond the questions and suggestions I will leave in "Questions", I identify what for me are two weaknesses of this paper:

1. The paper doesn't use "traditional" ML models (eg, SVM, random forests) directly on the flatten upper-triangle of the FC matrix as baselines. Based on my experience when running new deep learning models on FC connectivity, what I've found is that a simple traditional ML model (with a reasonable simple hyperparameter search) achieves comparable, if not better, metrics than many DL-based models on some datasets. Therefore, to evaluate the utility of this model, it would be important to see how BrainMAE compares to these more "traditional" ML models.
2. The paper is missing important metrics (e.g., sensitivity and specificity) in the evaluation section of the (binary) gender prediction. Given there's an appendix, and well-known limitations with accuracy and AUROC, I don't think it is a subjective comment to request these two other metrics in a paper that has a clear connection with the medical domain.


Two minor weaknesses that I've identified are:
1. Page 5 includes a part that is confusing in terms of readability. We have figure 2 mentioning SG-, DG-, and FDG- models, but right below in "BrainMAE Variants" only two are mentioned. Technically, they are not connected in the flow of the text, but visually they are together in the paper.
2. I understand that it's difficult to come up with these names, but in a work in which there is a key division between Static- and Dynamic-fMRI modelling, I find the static/dynamic naming for the graph attention component not the best choice as they seem to convey different meanings (one relates to use or not the entire fMRI timeseries, and the other relates to whether use the timeseries directly in the representation or one changed by an attention mechanism). Thus, I think it doesn't help with the clarity/readability of the paper to use these terms in two distinct contexts (unless I've missed something in the reasoning of the authors).

**Questions:**

1. Was there a particular reason for the authors to choose masked AEs in favour of other AEs? From the Introduction section it doesn't seem clear to me why this was the case. If the reason is just that they haven't been explored before in this field, I think it's a reasonable motivation (and results show it might have been a good choice). However, it would be good to say it why this was preferred to, for example, (V)AEs or other encoding variants as motivation for this new method.
2. I found the description on the "Static/Dynamic graphs attention" component confusing and unexpected. Calling it graph "attention" led me to think that some learning (for example in the form of some GNN) would happen, but from Section 2.1, it seems there's no learnable parameters, and what happens is that the representations are updated just once before being inputted to the TSE modules. If this is the case, and considering the important mentions to GNNs in the abstract, Introduction and Related Work sections, why haven't the authors decided to use a GNN instead of an attention transformation on a graph representation followed by a transformer? The authors do compare their work to other GNN-based baselines, but no baseline seems to have the same pipeline and pretraining procedure for a more direct comparison on the utility of GNNs. I do understand that at the end of the day there are methodological decisions that just need to be made, but this one is so close to GNNs that I do not understand why it hasn't been done, and would appreciate a clarification.
3. Why have self-loops been removed in the graph attention component? Self-loops are common standard in graph/GNN literature to keep the information from the initial node on the representation. Thus, it's not clear why the authors made this decision and why that's not at least in the ablation analysis.
4. The paper does a very good job explaining the different components of the model with regards to how it learns/creates the unsupervised brain representation. However, the paper gets a bit confusing (and even a bit unexpected) when in section 2.1 it is said in the "Autoencoder" component that the decoder is only used in pre-training phase (why is there such a division?); then, on section 3.1 when it is said that behaviour/demographic measurements are used (for what?); then, section 3.2.3 once again mentions that somehow we have a pretrained model for all datasets (why so much computational complexity?). Only in section 3.3.1 it is explained that the authors appended a task-specific linear head based on previous studies, and tables 1/2/etc also make it obvious that there are classification and regression tasks. For all these reasons, I hope it's clear why figure 1 (and consequently section 2) needs a quick mention to this task-specific head for better clarity/readability.
5. Why haven't the authors included the task-specific fMRI data from HCP, but decided to include the task-specific fMRI data from the NSD dataset?
6. Can the authors clarify what is the difference between the DG- and FDG-BrainMAE models? The FDG-BrainMAE is introduced as the model but without the "static graph mechanism". Isn't this basically the same as the DG-BrainMAE? My guess is that what the authors mean is that instead of the static graph mechanism in *the first layer* of DG-TSE, the FDG-BrainMAE's TSE module used only the dynamic graph representation across all blocks from the very beginning?
7. How are the folds created in the 5-fold cross-validation procedure? Is it in a stratified fashion independently for each downstream task? Are the authors careful to include subjects never seen before in training completely separated in the test sets?
8. Do the authors have any hypothesis on why the initial and final task blocks were particularly important for the model to make the final predictions?

---

> ### Author Response · Authors · 2023-11-18
> **Response to Reviewer DteR (1/3)**
>
> We greatly appreciate your excellent summary of the key aspects of our paper and careful examination of our paper. Thank you for the careful and insightful review! We've incorporated many of your suggestions. Below, we respond to each individual point from the review.
>
> > Q1: Missing comparison with traditional ML models.
>
> Thank you for raising this point. To address this concern, we analyzed the performance on the three downstream tasks with three classification models (SVM, Logistic Regression and Random Forrests), and three regression models (ordinary linear regression, ridge regression, Elastic Net) with details in $\textbf{Appendix G.4}$ and $\textbf{Tables 16-19}$ of our latest revised submission.
>
> Indeed, traditional ML models demonstrate performance levels comparable to those of baseline deep learning methods in Gender classification tasks. This observation can be attributed to the well-established understanding that FC matrix harbors significant information of human traits. However, for more complex regression tasks, such as task performance prediction on NSD dataset, which necessitate inferring intricate brain states from the dynamic of fMRI signals, simple ML models are less effective. In such scenarios, deep learning methods, endowed with the robust capability for representation learning, are able to achieve superior results. Nevertheless, our models continuously outperform these baseline methods by significant margin.
> > Q2: Missing metrics (e.g., sensitivity and specificity) in classification evaluation.
>
> Thank you for pointing this out. We acknowledge the importance of these metrics and have added them (sensitivity, specificity, and F1-score) to $\textbf{Tables 16-19}$ in the appendix of our revised submission. With these additional metrics, our method continues to significantly outperform other baseline models.
>
> > Q3: Readability in Page 5 regarding FDG-BrainMAE.
>
> Thank you for bringing this to our attention. To address this confusion, we will rename the FDG-BrainMAE to 'vanilla-BrainMAE' in our final revision. This reflects its use of TSE without embedding-informed graph attention, aligning with the typical use of 'vanilla' as model for comparison, which should improve clarity.
>
> > Q4: Confusion regarding the static/dynamic naming for the graph attention.
>
> Thank you for the suggestion. We acknowledge that the terms "static/dynamic" might be confusing in the context of fMRI, where "dynamic" typically refers to the time-evolving nature of fMRI signals. In our model, these terms were intended to denote spatially static or dynamic graphs for capturing transient representations. To clarify, we will rename 'dynamic' to 'adaptive' to better reflect the self-attention's adaptability to input signals in our final revision. Consequently, DG-TSE and DG-BrainMAE will be referred to as Adaptive-Graph TSE (AG-TSE) and Adaptive-Graph BrainMAE (AG-BrainMAE), respectively. This new terminology should reduce confusion with the convention.
>
> > Q5: Why choose masked AEs in favor of other AEs?
>
> Our motivation stems from the idea that masking a large set of ROIs and predicting their signals from unmasked ROIs can enhance the model's understanding of ROI relationships and improve ROI embedding learning. This approach is a key component of our method and has shown robust ROI representation learning. Additionally, in $\textbf{section 3.3.4's Ablation Study}$, we compared FDG-BrainMAE and FDG-BrainAE. Despite having the same architecture, they differ in pretraining methods: masked autoencoding for FDG-BrainMAE and standard autoencoding for FDG-BrainAE. Our results show that FDG-BrainMAE consistently surpasses FDG-BrainAE in all three downstream tasks by large margin, highlighting its effectiveness.

---

> > ### Author Response · Authors · 2023-11-18
> > **Response to Reviewer DteR (2/3)**
> >
> > > Q6: Learnable parameters in static/dynamic graph attention.
> >
> > We apologize for any confusion caused and thank you for raising this issue. We recognize that our original submission did not clearly explain the learnable parameters within the embedding-informed graph attention mechanism. To rectify this, we have thoroughly revised Section 2.1 in our upcoming submission to improve clarity.
> >
> > In the embedding-informed graph attention, the ROI embeddings undergo two separate learnable linear transformations before the attention computation.
> > $$
> > A_{ij} = s(W_qe_i, W_ke_j).
> > $$
> > where, $s: R^d\times R^d \longrightarrow R$ is the similarity measurement of two vectors, e.g., scaled dot-product and $e_i$, $e_j$ are the embedding vectors for node $i$ and $j$ respectively. The weight matrices $W_q, W_k\in R^{d\times d}$ are learnable and introduce asymmetry into $A$.
> >
> > This design reflects the asymmetric nature of functional relationships between ROIs in the brain. Therefore, this is the same as the attention performed in the transformer block when we replace the query and key with the ROI embeddings.
> >
> > The dynamic graph attention in our model is essentially a self-attention mechanism that also incorporates learnable linear transformations prior to the computation of attention.
> >
> >
> > > Q7: Why have self-loops been removed?
> >
> > Thanks for bringing up this question. This design is grounded in the nature of fMRI signals, where functionally relevant ROIs often show synchronized time-series signals during transient periods. Hence, the signal of each ROI can be inferred from other related ROIs. Removing self-loops in our model avoids the attention mechanism favoring its own node, encouraging it to aggregate signals from other relevant nodes. This approach is akin to a group voting system and helps reduce sensitivity to input noise.
> >
> > In Appendix G.2, we included an ablation study to assess downstream task performance by comparing SG-BrainMAE with and without self-loops. As indicated in Tables 12-14, the inclusion of self-loops results in slightly lower performance in HCP-3T and HCP-Aging tasks, and a more significant decline in the NSD task. This finding supports the benefits of our self-loop removal approach in the model design.
> >
> >
> > > Q8: Why is the decoder only used in pre-training phase in section 2.1 "Autoencoder"?
> >
> > Sorry for the confusion. To clarify, the decoder referenced in the 'Autoencoder' section is solely employed for reconstructing fMRI signals during the pretraining phase. It is not utilized in the finetuning of downstream tasks. As detailed in Appendix A.2 and shown in Figure 5, we integrate a task-specific linear head atop the CLS token from the encoder for downstream task finetuning.
> > > Q9: Where are behavior / demographic measurements used?
> >
> > For clarification, the behavior and demographic measurements are exclusively used in the downstream fine-tuning tasks on the HCP-3T dataset. These measurements are not incorporated into the pretraining phase of the models.
> >
> > > Q10: Why to have pre-trained models for all datasets?
> >
> > Thank you for your questions. To clarify, in Section 3.2.3, SG-BrainMAE is pretrained individually on each dataset, rather than collectively on all datasets. This approach was chosen to assess if the ROI embeddings, when pretrained separately on each dataset, would lead to convergent representations. As demonstrated in Figure 3 and detailed in Appendix D, we indeed observed convergence in the ROI embeddings.

---

> > > ### Author Response · Authors · 2023-11-18
> > > **Response to Reviewer DteR (3/3)**
> > >
> > > > Q11: Why figure 1 (and consequently section 2) needs a quick mention to this task-specific head for better clarity/readability.
> > >
> > > Sorry for the confusion. To clarify, Figure 1 introduces the decoder specifically for fMRI reconstruction, not for the task-specific heads used in our downstream tasks. The task-specific heads are described later in Section 3.3.1 and illustrated in Appendix Figure 5A.
> > >
> > >
> > >  > Q12: Why use the task-specific fMRI data from the NSD dataset but not from HCP?
> > >
> > > The NSD dataset contains direct measurement of memory task accuracy and behavioral arousal measurement such as response time. These metrics are ideal for testing whether the model is able to capture essential brain states from fluctuation of fMRI signals.
> > >
> > > In $\textbf{Appendix F}$ of our revised submission, we newly added a downstream task using the HCP task dataset, where the models are required to classify 20 mental states, with each representing distinct type of task and with duration of tens-of-seconds. Our model surpasses current state-of-the-art methods that are specifically designed for transient fMRI modeling[1].
> > >
> > > > Q13: What's the difference between the DG- and FDG-BrainMAE models?
> > >
> > > Your interpretation is absolutely right! The key distinction between DG-BrainMAE and FDG-BrainMAE lies in the use of the embedding-informed attention module. DG-BrainMAE incorporates this module in its first layer, whereas in FDG-BrainMAE, all layers within the TSE utilize self-attention.
> > >
> > > > Q14: How are the folds created in the 5-fold cross-validation procedure? Is it stratified? Are the authors careful to include subjects never seen before in training completely separated in the test sets?
> > >
> > > Our 5-fold cross-validation follows the standard approach, randomly assigning samples for testing in each fold, with the remaining divided into training and validation. In the three downstream tasks, we didn’t use stratified cross-validation nor separating traning and testing set based on subject identity.
> > >
> > > However, we recognize this concern and in our newly added HCP task for mental state classification (detailed in $\textbf{Appendix F}$), we conducted cross-validation based on subjects, ensuring no overlap between training and testing groups. Our proposed models consistently outperformed baseline methods.
> > >
> > > > Q15: Do the authors have any hypothesis on why the initial and final task blocks were particularly important for the model to make the final predictions?
> > >
> > > Thank you for raising this question. We hypothesize that the attention score might mirror specific brain states, such as transient arousal states critical for performing memory tasks. In $\textbf{section 3.3.5}$ of our revised paper, we explored this by analyzing the relationship between averaged behavioral brain arousal measurements (inverse response time) and averaged attention scores. We noticed that both metrics display similar patterns within and across task blocks, with high values at the start and end of these blocks. This pattern could indicate that our model inherently captures changes in transient arousal states, which may be vital for predicting memory task accuracy. Furthermore, we examined the correlation between fMRI representations and a well-defined brain arousal index, finding a strong correlation (details in $\textbf{Appendix E.2.2}$).
> > >
> > > ### References
> > > [1] A. Thomas, C. R ́e, and R. Poldrack. Self-supervised learning of brain dynamics from broad neuroimaging data. Advances in Neural Information Processing Systems, 35:21255–21269, 2022

---

> ### Comment · Reviewer_DteR · 2023-11-20
>
> I thank the authors for the really very good work on this new revision of the paper! Well done!
>
> I would suggest the authors to include most of the explanations they gave me in which I was very confused regarding parts of the paper. I believe particularly the one explaining the difference between the DG- and FDG-BrainMAE models would really benefit the readability of the paper, as the explanations are very simple but when I read the paper I believe it is not clear.
> I am a bit concerned about the fact that in the three downstream tasks, the authors did not use stratified cross-validation nor separated training and testing set based on subject identity. The fact they included that in the newly added HCP task-specific classification is reassuring, though unfortunately not completely "perfect" for the entire paper.
>
> Before I change my scores, I just have two follow-up questions:
> 1. How was the hyperparameter selection/search done for the newly added traditional ML models?
> 2. I think the choices of renaming "FDG-" to "vanilla-" and "Dynamic-" to "Adaptive-" are good ones! I was just wondering why haven't the authors included these changes yet given all the work they already did for this new version of the paper? Just lack of time?

---

> > ### Author Response · Authors · 2023-11-21
> > **Response to Reviewer DteR**
> >
> > Thank you for your positive feedback on our revised paper! We will add clearer explanations about the FDG-BrainMAE and clarify its differences from the proposed other MAE variants in our next revision. Your input is invaluable in improving our work! Below, we respond to the two questions from the review.
> >
> > > Q1: How was the hyperparameter selection/search done for the newly added traditional ML models?
> >
> > Thanks for bringing up the question. For hyperparameter selection in traditional ML models, we employed a cross-validated grid search approach, with specific ranges and increments for each model:
> >
> >  **Support Vector Machine (SVM)**: We varied the L2 regularization coefficient from 0.1 to 10, with an increment of 0.5.
> >
> > **Logistic Regression**: The L2 regularization coefficient was also tuned from 0.1 to 10, with an increment of 0.5.
> >
> > **Random Forest**: Three key parameters were tuned:
> > * Number of trees, ranging from 1 to 250 with increments of 50.
> > * Maximum depth of each tree, from 5 to 50 with increments of 10.
> > * Minimum samples required to split a node, from 5 to 100 with increments of 20.
> >
> > **Ordinary Linear Regression**: This model did not require hyperparameter tuning.
> >
> > **Ridge Regression**: The L2 regularization coefficient was tuned from 0 to 10, with an increment of 0.5.
> >
> > **Elastic Net Regression**:
> > * The coefficient of the L2 penalty (ridge regression component) was tuned from 0 to 10, with increments of 0.5.
> > * The coefficient of the L1 penalty (lasso regression component) was adjusted from 0 to 1, with increments of 0.2.
> >
> > For models requiring multiple hyperparameters, we trained each possible combination. The best-performing model was selected based on its performance on the validation set, using Mean Squared Error (MSE) for regression tasks and accuracy for classification tasks.
> >
> > > Q2: Why haven't the authors included renaming "FDG-" to "vanilla-" and "Dynamic-" to "Adaptive-" yet in this new version of the paper? Just lack of time?
> >
> > Thank you for your constructive feedback. The only concern we have at this stage is the potential confusion it may cause among other reviewers who have been engaged in the discussion process based on the existing terminology. However, we will implement these renamings in our final revision at the end of this review period, ensuring clarity and consistency in our paper.

---

> > > ### Comment · Reviewer_DteR · 2023-11-21
> > >
> > > Thanks for your further clarification. I'd say you should include those details about the hyperparameter search on the traditional ML models in appendix. As a suggestion, you might want to consider using random search in the future.
> > >
> > > I don't have any further questions, and I'll wait for the other reviewers before making a final decision on how to change my scores. Thanks once again for your time tackling my review.

---

> > > > ### Author Response · Authors · 2023-11-21
> > > > **Response to Reviewer DteR**
> > > >
> > > > Thank you for your additional feedback and advice! We have updated our latest submission to include the details of the hyperparameter search for traditional ML models in **Appendix G.4**. Based on your previous insightful comments, we have also added **Appendix Table 22** to clearly outline the model variants and emphasize their differences. The section on the **Ablation Study (3.3.4)** has been revised accordingly to reflect these updates. We highly appreciate your recommendation to consider random search in future studies, which we will certainly take into account for refining our model selection process.
> > > >
> > > > Thank you once again for your thorough review and valuable insights. Your input has been instrumental in enhancing the quality of our paper, and we are grateful for the time and effort you have invested in reviewing our work!

---

### Official Review · Reviewer_3Zps · 2023-10-31

**Soundness:** 3 good
**Presentation:** 2 fair
**Contribution:** 2 fair
**Rating:** 5
**Confidence:** 4

**Summary:**

This paper presents a method BrainMAE which uses Graph Neural Networks (GNNs) to analyze functional magnetic resonance imaging (fMRI) data for understanding brain network connectivity. BrainMAE employs a Transformer-based architecture to reconstruct masked segments of fMRI signals. The approach is validated by its ability to encode functional connectivity in a reproducible manner across different datasets and downstream tasks.

**Strengths:**

1. The paper is well-written and easy to follow.
2. The paper is well structured and has a smooth and concrete flow.
3. The experimental part in this paper is abundant.

**Weaknesses:**

1. While the BrainMAE model is presented as a fresh approach to fMRI data, it's hard to overlook the fact that all Transformer Layers, Masked Autoencoders (MAE) and Dynamic Graph Neural Networks are somewhat antiquated techniques. These methods are not only dated in the broader machine learning landscape but have also lost their novelty in computational neuroscience applications [1,2,3]. Frankly, the proposed method comes across as a 'engineering model'.
2. No theoretical guarantees of the proposed method. The BrainMAE seems to be too heuristic.
3. You mention your method is 'interpretable' in the title. However, I don't figure out the true interpretability of the method since BrainMAE has no scientific inductive bias incorporated. This is just a stacking of deep-learning-based techniques, which make the model highly black-boxed. Please refer to the Questions Section for my further concerns.


[1] MAEEG: Masked Auto-encoder for EEG Representation Learning.
[2] Pooling regularized graph neural network for fmri biomarker analysis.
[3] Graph neural network for interpreting task-fmri biomarkers.

**Questions:**

For your Subsection 3.3.5, Representation and Interpretation Analysis. I hold the point that analyzing the representations of fMRI using principal component analysis (PCA) and self-attention scores are too post hoc  and lacks scientific insights. In fact, the representations extracted by BrainMAE and attention layers are still black-box.

**Details Of Ethics Concerns:**

I have no concerns at this step.

---

> ### Author Response · Authors · 2023-11-18
> **Response to Reviewer 3Zps (1/2)**
>
> Thank you for bringing up the concerns and for your insightful feedback. We apologize for the lack of clarity in our initial description. In response, we have revised relevant sections of the paper to more clearly highlight the scientific inductive biases. We've additionally conducted comprehensive analyses of the intermediate model generated representations and aiming to more directly connect them with scientific insights. Below, we respond to each individual point from the review.
> > Q1: The proposed methods have lost their novelty in computational neuroscience applications and come across as a 'engineering model'.
>
> Thank you for raising this concern. ML methods are indeed not new in computational neuroscience, but there are inherent challenges with fMRI, a critical tool for brain research. Unlike other signals such as EEG, fMRI's unique temporal resolution and spatial coverage require distinct modeling approaches.
>
> Many existing ML models for fMRI, including the two papers you cited, primarily compute functional connectivity matrices and often overlook the importance of dynamic brain state changes. These dynamics carry rich information, proved to be a major contribution to the FC, and relates to a variety of brain function, showing promise for understanding how the brain actually works[1].
>
> On the other hand, deep learning methods that consider fMRI's dynamic nature frequently underperform compared to fixed-FC approaches, partly due to fMRI's inherent noise. Moreover, several dynamic-FC approaches employ extended time windows (30-60 seconds) to manage noise and spurious connections, but this compromises their sensitivity to brief, transient brain state changes that typically happen within 10-20 seconds[1][2].
>
> Our model addresses these challenges by introducing a novel ROI embedding-informed attention mechanism and a uniquely designed MAE training scheme. This approach not only overcomes the limitations seen in previous dynamic-FC models but also consistently outperforms state-of-the-art methods. The significance of these new components is evident in our ablation study, where their removal results in a substantial performance decrease.
>
> Because of these specifically designed features, our model holds promise for use in neuroscience and medical research. It's particularly suited for unraveling how transient fMRI fluctuations contribute to overall brain representations in various circumstances, aiding in the study of brain traits, states, and diseases.
>
>
> > Q2: No theoretical guarantees of the proposed method. The BrainMAE seems to be too heuristic.
>
> We acknowledge that, at this stage, our approach is more heuristic and grounded in empirical insights from neuroscience and machine learning. The absence of formal theoretical guarantees is a valid point, and we view it as an important area for our future research. Our current focus has been on leveraging intuitive understandings of brain functionality and the nature of fMRI signals to develop a model that shows promising empirical results. We aim to complement these findings with more theoretical analysis in our subsequent work, and we believe this will strengthen the method's scientific contribution.

---

> > ### Author Response · Authors · 2023-11-18
> > **Response to Reviewer 3Zps (2/2)**
> >
> > > Q3: Can't figure out the true interpretability of the method since BrainMAE has no scientific inductive bias incorporated.
> >
> > Thank you for highlighting this point. We realize our initial submission lacked clarity on this aspect, and we've addressed it in our revised submission with detailed explanations of the scientific inductive biases we've incorporated:
> >
> > 1. ROI embedding (section 2.1): Each ROI has its unique inherent characteristics and we resemble it as the words in the NLP context by assigning each ROI with learnable vector. Section 3.2.3 demonstrates that these ROI embeddings are informative and consistently converge across different datasets.
> >
> > 2. ROI embedding-informed attention module (section 2.1):
> >
> > * As brain ROI activities are intrinsically governed by both structural and functional networks, we consider building adjacency matrix based on ROI embeddings to represent such functional networks.
> >
> > * Considering the asymmetric flow of information between ROIs, we apply two separate learnable linear transformations to the embeddings before attention computation, introducing asymmetry in the adjacency matrix.
> >
> > * Based on the synchronized nature of fMRI signals among functionally relevant ROIs, we eliminate self-loops in the attention mechanism. This design helps prevent bias towards individual nodes and enhances feature learning by reducing susceptibility to input noise, similar to a group voting process.
> >
> > 3. Masked Autoencoding (section 2.2)
> >
> > * We segment the fMRI using 15-second time window aligning with the typical duration of transient events in fMRI data.
> >
> > * As functionally relevant brain regions show synchronization in performing certain brain functions, we mask large portion of fMRI ROI signals and optimize the model to reconstruct the masked signal from unmasked signal, which is useful to discover underlying functional relevance between ROIs thus improving the learning of ROI embeddings.
> >
> > > Q4: Representation of fMRI using PCA and self-attention scores are too post hoc and lack scientific insights. In fact, the representation extracted by BrainMAE and attention layers are still black box.
> >
> > Thanks for pointing out. To address this concern, we performed three additional representation analyses.
> >
> > 1. In Appendix E.1, we apply the pretrained SG-BrainMAE on the never-seen NSD dataset and extract the representation for each fMRI run. Surprisingly, in tSNE analysis, we found the subject identity is clearly separated from each other, suggesting the pretrained model is able to discern unseen subject identity purely from their fMRI signals.
> >
> > 2. In Appendix E.2, same as the above setup, within each subject, in tSNE analysis, the fMRI type (task/rest) can be discernable for most of the subjects, suggesting the pretrained model's ability to recognize different brain states without specifically trained to.
> >
> > 3. In Appendix D.3, we categorized subjects from the HCP-Aging dataset into three age groups: young, middle, and old. We pre-train SG-BrainMAE individually for each group and study the age effects on the learned ROI embeddings. The modular structure of the network constructed based on the embeddings, quantified by modularity, indicates a reduction with age increasing, aligning with the established findings in neuroscience research.
> >
> > Additionally, regarding the self-attention analysis in section 3.3.5 (Figure 4), we conducted further assessments.  Interestingly, we found the attention score correlates to the behavioral arousal measurements, the inverse reaction time, and across- and within-task block, suggesting the model is aware of the change of transient brain state.
> >
> >
> >
> >
> >
> > ### References
> > [1] T. Bolt, J. S. Nomi, D. Bzdok, J. A. Salas, C. Chang, B. Thomas Yeo, L. Q. Uddin, and S. D. Keilholz. A parsimonious description of global functional brain organization in three spatiotemporal patterns. Nature neuroscience, 25(8):1093–1103, 2022.
> >
> > [2] J. M. Shine, P. G. Bissett, P. T. Bell, O. Koyejo, J. H. Balsters, K. J. Gorgolewski, C. A. Moodie, and R. A. Poldrack. The dynamics of functional brain networks: integrated network states during cognitive task performance. Neuron, 92(2):544–554, 2016

---

> ### Comment · Reviewer_3Zps · 2023-11-22
>
> I thank the authors for the revised manuscript and the additional experiments. These empirical results have addressed some of my concerns and enhance the overall quality of this work. I have raised my score accordingly.

---

> > ### Author Response · Authors · 2023-11-22
> > **Response to Reviewer 3Zps**
> >
> > We are pleased to hear that the revision has addressed some of your concerns. Your feedback was crucial in initiating these enhancements, and we are also grateful for the increase in your evaluation score, which is highly encouraging to us. Thank you once again for your review and suggestions.

---

### Official Review · Reviewer_8czh · 2023-11-08

**Soundness:** 3 good
**Presentation:** 2 fair
**Contribution:** 3 good
**Rating:** 6
**Confidence:** 3

**Summary:**

In this paper, the authors developed Brain Masked Auto-Encoder (BrainMAE) for representation learning on brain fMRI data. The authors combined two newly-defined graph attention modules with MAE to obtain the latent representations for downstream analysis. The method is evaluated on various datasets to demonstrate the robustness and functional relevance of the obtained representations, and compared with several existing baselines to demonstrate superior performances in downstream tasks.

**Strengths:**

The proposed method is original and serves as one of the first attempts to use MAE in the field of brain imaging representation learning. Quality of the evaluations is good and convincing: the learned representations show good consistency between different datasets and good functional relevance.

**Weaknesses:**

The writing is generally good, but the flow of the paper can be improved as certain parts can be hard to follow. For example, the notations in Section 2 is not listed clear enough: "node features" are described in equation (3) before the actual input fMRI segment X is formally introduced in Section 2.2. Some of the statements may not be very well supported, mostly regarding the dynamic graph attention. I have listed my questions below.

**Questions:**

1. What exactly is shown in Figure 3A? Is this a representative t-SNE plot for one subject? Or did the authors take the mean ROI embeddings across all subjects and plotted t-SNE afterwards? Or something else?
2. From the evaluations in all tables, it seems that SG-BrainMAE consistently outperforms DG-BrainMAE. If that is the case, I can't see the reasoning of introducing dynamic graph attention modules. I hope the authors could expand a bit on that. Also, why is it called "dynamic"?

---

> ### Author Response · Authors · 2023-11-18
> **Response to Reviewer 8czh**
>
> Thank you very much for your acknowledgment of our work’s quality and your valuable suggestions, your questions provided us with a chance to enhance our paper and improve the flow of our writing. We sincerely appreciate your time in reading the paper, and our point-to-point responses to your comments are given below.
> > Q1: Node features are described before formally defined.
>
> Thank you for pointing this out. In our revised submission, for the definition of the node features, we added sentence as follows: "Within this graph, each ROI is considered a node, with its node feature represented by the ROI's time-series signal of length $\tau$ or $x\in R^{\tau}$".
> > Q2: Some of the statements may not be very well supported, mostly regarding dynamic graph attention.
>
> Sorry for the confusion and thank you for sharing your concern. We recognize the potential misunderstanding about 'dynamic graph attention,' which is simply the self-attention used in transformer models and not a major novelty of our work. To improve readability, we've omitted this description in our revised submission. Additionally, we've extensively revised Section 2.1 to more clearly convey our motivation, the novelty, and the inductive biases considered in our model's development. Feel free to suggest more comments about them and we would be happy to address them further.
> > Q3: How to get t-SNE plot in Figure 3A? Is this for one subject? Or the t-SNE of mean ROI embedding across all subjects?
>
> Apologies for the confusion. The ROI embeddings are not derived from a single subject or an average across subjects. They are, in fact, model weights learned globally using data from all subjects in the dataset, akin to word embeddings in NLP. These pretrained ROI embeddings represent the rich ROI-related information "summarized" from all the training samples.
> > Q4: SG-BrainMAE consistently outperforms DG-BrainMAE.
>
> Indeed SG-BrainMAE consistently outperforms DG-BrainMAE. However, In $\textbf{Appendix F}$ of our revised submission, we newly add a downstream task using the HCP task dataset. Here, models classify between 20 mental states, each lasting tens of seconds. In this context, both SG-BrainMAE and DG-BrainMAE outperform current state-of-the-art models[1], with DG-BrainMAE showing a slightly better performance relative to SG-BrainMAE. This suggests that incorporating dynamic components can be beneficial for detecting transient state changes in fMRI data over short timescales.
>
> In tasks requiring inference of 'static' measurements (such as behaviors, gender, or overall brain states) from longer-duration fMRI data, SG-BrainMAE's consistent superiority might imply that dynamic components in attention are less critical in these scenarios.
> > Q5: What is the reasoning for introducing dynamic graph attention modules? Also, why is it called "dynamic"?
>
> Thanks for bringing up this question. Dynamic graph attention essentially is self-attention, where the attention is directly derived from the input node feature without being constrained by the ROI embeddings. Therefore, the attention is "dynamic" and conditioned on the temporal fluctuation of ROI signals, thus representing the brain dynamic reconfiguration of ROIs connectivity. DG-BrainMAE, introduced as a complement to SG-BrainMAE, focuses on this dynamic aspect, unlike SG-BrainMAE, which assumes a static functional brain organization.
>
> Our experiments show that SG-BrainMAE is more effective in predicting static measurements like behaviors and gender. Conversely, DG-BrainMAE excels in identifying transient mental state fluctuations ($\textbf{Appendix F}$ of our revised submission). These results provides support for our desgin principles.
>
> ### References
>
> [1] A. Thomas, C. R ́e, and R. Poldrack. Self-supervised learning of brain dynamics from broad neuroimaging data. Advances in Neural Information Processing Systems, 35:21255–21269, 2022

---

### Author Response · Authors · 2023-11-22
**General Response**

We would like to sincerely appreciate the reviewers for their insightful and constructive suggestions. Every comment, whether positive or critical, has greatly helped us to further polish the paper. Here's a summary of the key changes we've made in the latest revision. For easy reference, we also indicated all changes by the reviewer who suggested them:

1. Readability: Renamed the FDG-BrainMAE to "vanilla-BrainMAE" to improve clarity (DteR).

2. Readability: Renamed DG-TSE and DG-BrainMAE to Adaptive-Graph TSE (AG-TSE) and Adaptive-Graph BrainMAE (AG-BrainMAE), respectively, to better reflect the adaptability of the attention to input signals, to distinguish from the term 'dynamic' in the context of fMRI which usually reflect time-evolving nature of the signals (DteR).

3. Readability: Thoroughly revised section 2.1 to enhance clarity and readability (8czh, DteR, nn5J).

4. ROI embedding interpretation: Added new experiments analyzing age effects on ROI embeddings with modularity analysis in Appendix D.3. (3Zps, nn5J)

5. Representation analysis: Added tSNE analysis of fMRI representations from previously unseen NSD dataset using the pretrained BrainMAE, revealing insights into subject identity (Appendix E.1) and TAST/REST states (Appendix E.2.1). (3Zps)

6. Attention score interpretation: Added an analysis exploring the relationship between attention scores and behavioral arousal measurements, uncovering a close relationship between the two (Figure 4). (DteR)

7. Transfer learning: Added a new HCP transient mental state decoding experiment in Appendix E, demonstrating that BrainMAEs outperform several state-of-the-art self-supervised learning models [1] designed for transient state modeling. (nn5J, 8czh)

8. Ablation study: Introduced SG-BrainMAE (SL) model, which includes self-loops in the embedding-informed graph attention, and added experiments in Appendix G.2 to validate our design choice of self-loop removal. (DteR)

9. Ablation study: Introduced model PosSG-BrainMAE, which replaces ROI embeddings with fixed position embeddings, and added experiments in Appendix G.1 to validate valuable information beyond position contained in the ROI embeddings. (nn5J)

10. Comparison with TFF: Added a performance comparison with the self-supervised model TFF [2] in HCP gender prediction in Appendix G.3. (nn5J)

11. Comparison with traditional ML models: Added a comparison with various traditional ML models to evaluate the performance gains of deep learning-based models in several downstream tasks in Appendix G.4. (DteR)

We once again thank each reviewer for the time and effort invested in providing their thorough review and valuable insights.


### Reference

[1] A. Thomas, C. R ́e, and R. Poldrack. Self-supervised learning of brain dynamics from broad neuroimaging data. Advances in Neural Information Processing Systems, 35:21255–21269, 2022.

[2] I. Malkiel, G. Rosenman, L. Wolf, and T. Hendler. Self-supervised transformers for fmri representation. In International Conference on Medical Imaging with Deep Learning, pages 895–913. PMLR, 2022.

---

### Meta-Review · Area_Chair_zASS · 2023-12-09

**Metareview:**

This paper introduces BrainMAE, a novel method for fMRI data analysis. BrainMAE combines graph attention modules with masked auto-encoding to learn robust and functionally relevant representations of brain activity. The method's effectiveness is demonstrated through its strong performance on various datasets and downstream tasks, showcasing its ability to capture both static and dynamic functional connectivity in the brain. Furthermore, BrainMAE's interpretability offers valuable insights into the decision-making process of the model, making it a promising tool for advancing our understanding of brain networks.

While BrainMAE presents a novel approach to fMRI analysis, it suffers from weaknesses in model design, interpretability, evaluation, and writing. Specifically, there is a lack of comparison with existing self-supervised learning models for fMRI analysis. There are also conflicting conclusions about inherent noise and model performance. The novelty of employing well-established techniques like Transformers and Masked Autoencoders is questionable. The paper misses theoretical guarantees and scientific inductive bias, hindering interpretability. Finally, the flow of the paper needs improvement, particularly in Section 2 where notations are introduced prematurely. Certain statements lack sufficient support and require clearer justification, especially regarding dynamic graph attention. Addressing these concerns through thorough comparison with relevant baselines, appropriate evaluation metrics, clear explanations and justifications, and potentially incorporating theoretical foundations would significantly strengthen the paper's contribution to the field of computational neuroscience and make it a stronger publication.

**Justification For Why Not Higher Score:**

While BrainMAE presents a novel approach to fMRI analysis, it suffers from weaknesses in model design, interpretability, evaluation, and writing. Specifically, there is a lack of comparison with existing self-supervised learning models for fMRI analysis. There are also conflicting conclusions about inherent noise and model performance. The novelty of employing well-established techniques like Transformers and Masked Autoencoders is questionable. The paper misses theoretical guarantees and scientific inductive bias, hindering interpretability. Finally, the flow of the paper needs improvement, particularly in Section 2 where notations are introduced prematurely. Certain statements lack sufficient support and require clearer justification, especially regarding dynamic graph attention. Addressing these concerns through thorough comparison with relevant baselines, appropriate evaluation metrics, clear explanations and justifications, and potentially incorporating theoretical foundations would significantly strengthen the paper's contribution to the field of computational neuroscience and make it a stronger publication.

**Justification For Why Not Lower Score:**

N/A

---

### Decision · Program_Chairs · 2024-01-16

Reject